# Understanding the Design Space and Cross-Modality Transfer for Vision-Language Models

## Abstract

The training of multimodal models involves many design choices, such as the underlying modality-specific tokenizers, fusion mechanisms, and strategies for freezing model layers during different training stages. However, the individual impact of these decisions on downstream multimodal performance remains poorly understood due to the diversity of current practices. In this paper, we systematically investigate how choices in image tokenization, architectural design, and layer-freezing strategies affect the training and cross-modal generalization of vision-language models (VLMs). We systematically explore a design space comprising six image tokenizers, three VLM architectural variants, and various parameter-freezing strategies. To further probe cross-modality transfer, we introduce three new synthetic datasets, which we use to evaluate our pretrained models. Our experiments reveal several key trends. **(i)** Image tokenizers trained with text-aware objectives are crucial for strong VLM performance, outperforming those trained without such objectives on both in-domain and out-of-domain tasks. **(ii)** Architectures that explicitly separate modalities such as the Mixture-of-Transformers fusion architecture, along with training recipes that preserve the more general textual knowledge and reasoning of the base language model, generalize well to out-of-domain tasks. **(iii)** Cross-modality transfer is heavily dependent on representational alignment between the text and images; in our synthetic setting, image-to-text transfer is comparatively strong, whereas there was little text-to-image transfer.

## 1 Introduction

Vision-language models (VLMs) are frequently built by combining a pretrained large language model (LLM) with an image tokenizer via a fusion architecture that integrates image and text representations. Here, we use the term *tokenizer* to refer to any module that processes raw modality inputs (such as images or text) into a sequence of discrete or continuous tokens for downstream modeling. This encompasses both discretization approaches such as VQVAE (van den Oord et al., 2017), and conventional encoders such as CLIP (Radford et al., 2021) and SigLIP (Zhai et al., 2023), which generate token-like embeddings.

Despite rapid progress, the effects of architectural and training choices, such as how to align and fuse modality-specific representations, remain unclear, due to the proliferation of fusion architectures (e.g., joint autoregressive decoders (Liu et al., 2023; Deitke et al., 2024; Bai et al., 2025; Du et al., 2025; Zhu et al., 2025), cross-attention models (Alayrac et al., 2022; Grattafiori et al., 2024; Dai et al., 2024) and mixtures-of-transformers (Liang et al., 2025; Shi et al., 2025b; Deng et al., 2025)), tokenization schemes (Tschannen et al., 2025; Fini et al., 2024; Oquab et al., 2023; Yu et al., 2024; Tian et al., 2024; Bachmann et al., 2025; Miwa et al., 2025), and multi-stage training recipes with varied layer freezing. This diversity of approaches makes it challenging to disentangle how each design choice impacts the performance and generalization behavior of VLMs.

Understanding how design and training strategies of multimodal models enable cross-modality transfer is particularly important. Effective transfer allows models to develop reasoning and understanding that may be more naturally expressed in one modality than another (for example, physical

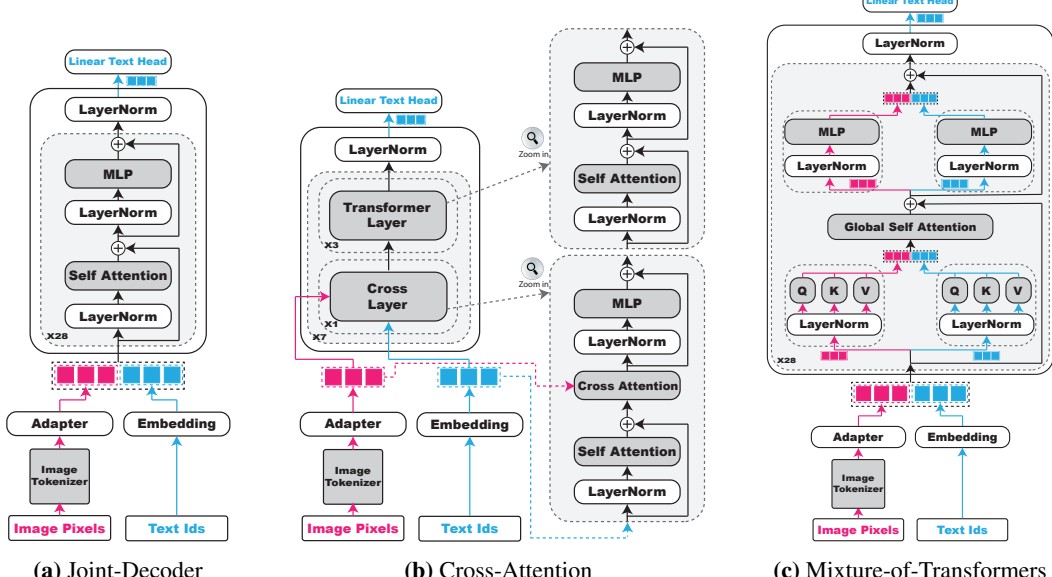

**Figure 1: Different multimodal fusion architectures:** (a) Joint-Decoder, (b) Cross-Attention, (c) Mixture-of-Transformers.

reasoning may be more apparent in vision than in text). Furthermore, robust cross-modality transfer enables models to leverage alternative data sources, which is increasingly valuable as high-quality text data becomes scarce due to the growing scale of LLM training.

In this work, we systematically investigate the influence of fusion architectures, image tokenizers, and training recipes on VLM performance. We instantiate and evaluate over 50 VLM configurations, all built on a Qwen3-0.6B (Yang et al., 2025) backbone, by ablating across six image tokenizers, three architectural variants, and various parameter-freezing strategies, trained on a mix of image–caption and visual question answering (VQA) data. We assess in-domain and out-of-domain performance using a suite of VQA benchmarks to discover trade-offs between architectural and training choices. To further explore cross-modality transfer, we also introduce three new synthetic datasets designed for evaluating cross-modality generalization.

Our main findings are: **(i)** Image tokenizers trained with text-aware objectives are crucial for strong VLM performance, outperforming those trained without such objectives on both in-domain and out-of-domain tasks. **(ii)** Architectures that explicitly separate modalities such as the Mixture-of-Transformers fusion architecture, along with training recipes that preserve the more general textual knowledge and reasoning of the base language model, generalize well to out-of-domain tasks. **(iii)** Cross-modality transfer is heavily dependent on representational alignment between the text and images; in our synthetic setting, image-to-text transfer is comparatively strong, whereas there was little text-to-image transfer.

## 2 EXPERIMENTAL SETTINGS

### 2.1 ARCHITECTURES

We study three fusion architectures for vision-language modeling, illustrated in Figure 1: Joint-Decoder, Cross-Attention, and Mixture-of-Transformers (MoT).

**Joint-Decoder:** In the Joint-Decoder architecture (Deitke et al., 2024; Bai et al., 2025; Du et al., 2025; Zhu et al., 2025), image and text tokens are concatenated and then fed into a shared multimodal transformer decoder (Figure 1a).

**Cross-Attention:** In the Cross-Attention architecture, cross-attention layers allow text tokens to attend to image token representations (Figure 1b), following architectures like Flamingo (Alayrac et al., 2022) and Llama 3-V (Grattafiori et al., 2024). Design variants exist in how and where visual

information is injected. For example, NVLM (Dai et al., 2024) explores different placements and use of special visual tokens for cross-modal reasoning.

**Mixture-of-Transformers (MoT):** For the Mixture-of-Transformers architecture (Liang et al., 2025; Shi et al., 2025b; Deng et al., 2025), the model processes each modality through its own stack of transformer layers, referred to as *modality-transformers*, each with separate query-key-value (QKV) matrices and feed-forward networks. At every layer, tokens are first routed through their respective modality-transformer for QKV computation, then mixed globally via multimodal self-attention, and finally passed back through that modality's feed-forward sublayer (Figure 1c). This doubles the parameter count of the fusion layers relative to the analogous Joint-Decoder architecture, but, as shown in (Liang et al., 2025), the overall floating-point operations (FLOPs) per forward pass remain comparable.

Given a pretrained LLM of a fixed size, we investigate the best fusion architecture to build a VLM from and experiment on three main ones: 1) Joint-Decoder, 2) Cross-Attention, and 3) Mixture-of-Transformers. We choose to build from Qwen3-0.6B, which contains 28 transformer layers, as the language backbone. While Qwen3-0.6B ties its text embedding and output head, we untie them for our models. In the **Joint-Decoder** architecture, these 28 layers are repurposed as a single, shared multimodal decoder initialized with the pretrained LLM weights. For the **Cross-Attention** models, we follow the design of Llama 3-V (Grattafiori et al., 2024) and interleave a cross-attention layer every four layers within the backbone, adding a total of seven new cross-attention layers. Finally, the **Mixture-of-Transformers (MoT)** architecture, following Shi et al. (2025b); Deng et al. (2025), creates two parallel modality-transformers, each containing a full copy of the 28 transformer layers, both of which are initialized with the original Qwen3-0.6B weights.

## 2.2 IMAGE TOKENIZERS AND ADAPTERS

We experiment with a range of image tokenizers, each trained with different objectives. In particular, we distinguish between *text-aware tokenizers*, which are trained with some form of textual training, and *text-blind tokenizers*, which contain no text in their training data.

- **Text-aware tokenizers:**
    - **CLIP** (Radford et al., 2021): Trained via contrastive learning to align image and text embeddings.
    - **SigLIP 2** (Tschannen et al., 2025): Trained using a combination of contrastive learning with a sigmoid loss (Zhai et al., 2023), an autoregressive captioning loss (Wan et al., 2024), a masked image prediction loss (He et al., 2022), and self-distillation (Naeem et al., 2025; Maninis et al., 2025).
    - **AIMv2** (Fini et al., 2024): Trained with next patch prediction (for images) and an autoregressive captioning loss (for text).
- **Text-blind tokenizers:**
    - **DINOv3** (Siméoni et al., 2025): Trained to extract image features via a DINO image-level loss (Caron et al., 2021), an iBOT latent reconstruction loss (Zhou et al., 2022), a Koleo regularizer (Sablayrolles et al., 2019), and a Gram anchoring objective.
    - **TiTok** (Yu et al., 2024): Trained to encode images into one-dimensional latent token sequences by reconstructing ground-truth two-dimensional latents.
    - **VAR** (Tian et al., 2024): Trained to autoregressively reconstruct images via multi-scale token maps.

To match tokens to fusion architectures, we include lightweight adapter modules. For Joint-Decoder and MoT, continuous tokenizers (CLIP, AIMv2, SigLIP 2, DINOv3) are projected using a two-layer MLP, while discrete tokenizers (TiTok, VAR) use an embedding layer. In the Cross-Attention architecture, adapters follow Llama 3-V, aligning dimensions for image tokens (using an embedding layer for discrete tokenizers and dimensionality matching for continuous ones).

## 2.3 TRAINING SETUP

We train each of our models in three stages: 1) a pretraining stage; 2) a VQA fine-tuning stage; 3) and a reasoning-transfer stage. The hyperparameters for all three training stages can be found in Appendix D.

**Stage 1 (Pretraining):** We pretrain models for caption generation on COYO-700M (Byeon et al., 2022) to align the image tokenizer and any uninitialized model weights, improving the representations available to the LLM layers. The language model layers are kept frozen in this stage. For each model with a continuous image tokenizer, we run both frozen and unfrozen variants to enable downstream comparison. Discrete image tokenizers remain frozen throughout, avoiding the need for a straight-through estimator.

During pretraining, we only train the Joint-Decoder and Cross-Attention models. For MoT models, we follow Shi et al. (2025b); Deng et al. (2025) and initialize weights from a trained Joint-Decoder checkpoint by transferring the adapter and image tokenizer weights, and copying the original Qwen3 weights into both modality-transformers.

**Stage 2 (Fine-tuning):** We fine-tune each pretrained checkpoint on a combination of COCO-Captions (Lin et al., 2014), VQAv2 (Goyal et al., 2017), DocVQA (Mathew et al., 2021), TextVQA (Singh et al., 2019), and ChartQA (Masry et al., 2022). We systematically ablate whether the image tokenizer or LLM layers are frozen at this stage, such that unfrozen layers during pretraining remain unfrozen for fine-tuning. Models are evaluated after each epoch on the validation splits of VQAv2, A-OKVQA (Schwenk et al., 2022), DocVQA, TextVQA, and ChartQA. The checkpoint with the highest mean validation accuracy across VQA datasets is selected for further evaluation.

**Stage 3 (Reasoning-Transfer):** We further train our models on three synthetic datasets, detailed in Section 4 and Appendix C. Each dataset pairs an image with an equivalent text description, allowing for both image-based and comparable text-only training runs. We evaluate on in-distribution and out-of-distribution tasks in both modalities to quantify cross-modality transfer.

### 2.4 EVALUATION PROTOCOL

To assess model capabilities, we evaluate on a combination of standard academic vision-language benchmarks and our own synthetic datasets. These benchmarks measure both general visual understanding and the ability to transfer knowledge across domains and modalities. We group academic benchmarks into in-domain and out-of-domain, where in-domain benchmarks test short answer questions with similar topics to our training data, and out-of-domain benchmarks not only cover novel topics but also feature visual multiple-choice questions, a format our models were not exposed to during training.

For academic evaluations, our in-domain suite tests a range of capabilities, including general VQA with **VQAv2** (Goyal et al., 2017), knowledge-based reasoning with **A-OKVQA** (Schwenk et al., 2022), specialized understanding of documents (**DocVQA** (Mathew et al., 2021)), text in images (**TextVQA** (Singh et al., 2019)), and charts (**ChartQA** (Masry et al., 2022)). To assess out-of-domain generalization, we use **MathVista** (Lu et al., 2024b) for mathematical reasoning, **RealWorldQA** (xAI, 2024) for robustness to novel image distributions, and the multi-task benchmark **MMTBench** (Ying et al., 2024).

## 3 UNDERSTANDING THE DESIGN OF MULTIMODAL ARCHITECTURES

### 3.1 EFFECT OF IMAGE TOKENIZER ON IN-DOMAIN PERFORMANCE

Table 1 presents the accuracies for different image tokenizers within the Joint-Decoder architecture, with the image tokenizer frozen to normalize for the trainability of the discrete tokenizers TiTok and VAR. The results show that image tokenizers trained with text-aware objectives (CLIP, AIMv2, SigLIP 2) generally outperform those with text-blind objectives (DINOv3, TiTok, VAR) for both in-domain and out-of-domain tasks. In particular, if we look at DINOv3, AIMv2, and SigLIP 2, all of which are recent tokenizers with similar architectures and trained on similar scales of data, we find that the text-aware tokenizers AIMv2 and SigLIP 2 outperform the text-blind tokenizer DINOv3 by an average of around 7-10 points on in-domain tasks. This general trend holds true across the cross-attention and MoT architectures and is consistent regardless of the layer freezing strategy. Full evaluation results for all architectures and tokenizers are available in Appendix A.

**Takeaway 1.** Image tokenizers trained with text-aware objectives are crucial for strong VLM performance, outperforming those trained with text-blind objectives, especially on in-domain tasks.

**Table 1: Evaluation results (accuracy in %) on the Joint-Decoder architecture with frozen image tokenizer.** The table compares results across various image tokenizers while ablating whether the language model (**Lang**) is frozen (❄) or unfrozen (🔥) during Stage 2. The highest score in each column is bolded.

| Tokenizer | Lang | VQAv2 test-dev | A-OKVQA val | ChartQA test | TextVQA val | DocVQA test | Average In-domain | Average Out-of-domain |
|---|---|---|---|---|---|---|---|---|
| CLIP | ❄ | 49.8 | 22.1 | 12.2 | 20.1 | 12.1 | 23.2 | 30.5 |
| CLIP | 🔥 | 66.7 | 41.6 | 20.8 | 31.9 | 20.7 | 36.3 | 32.9 |
| AIMv2 | ❄ | 62.5 | 30.3 | 19.5 | 33.6 | 17.9 | 32.8 | 32.4 |
| AIMv2 | 🔥 | **75.3** | **49.5** | **30.6** | 43.2 | 25.2 | **44.7** | **34.5** |
| SigLIP 2 | ❄ | 55.9 | 24.0 | 15.7 | 30.0 | 17.0 | 28.5 | 31.7 |
| SigLIP 2 | 🔥 | 74.8 | 47.2 | 28.8 | **43.6** | **26.6** | 44.2 | 32.8 |
| DINOv3 | ❄ | 53.6 | 22.8 | 9.6 | 12.7 | 9.6 | 21.7 | 30.2 |
| DINOv3 | 🔥 | 71.5 | 45.7 | 19.0 | 20.5 | 15.8 | 34.5 | 29.7 |
| TiTok | ❄ | 3.2 | 0.2 | 5.1 | 1.0 | 3.0 | 2.5 | 26.7 |
| TiTok | 🔥 | 43.1 | 26.0 | 13.9 | 11.8 | 11.8 | 21.3 | 27.1 |
| VAR | ❄ | 30.3 | 2.0 | 9.2 | 4.9 | 6.0 | 10.5 | 28.1 |
| VAR | 🔥 | 46.5 | 27.7 | 13.7 | 11.9 | 11.9 | 22.3 | 22.4 |

## 3.2 VARYING FROZEN LAYERS AND ARCHITECTURAL CHOICES

**Table 2: Average in-domain and out-of-domain performance (accuracy in %) across different fusion architectures and layer freezing strategies.** Scores are averaged over all text-aware image tokenizers (CLIP, AIMv2, SigLIP 2). The table ablates the freezing status, Frozen (❄) or Unfrozen (🔥), of the image tokenizer and language model during Stage 1 (pretraining) and Stage 2 (fine-tuning). The highest score in each column is bolded.

| Image Stage 1 | Image Stage 2 | Language Stage 2 | Joint-Decoder In-domain | Joint-Decoder Out-of-domain | Joint-Decoder Overall | Cross-Attention In-domain | Cross-Attention Out-of-domain | Cross-Attention Overall | MoT In-domain | MoT Out-of-domain | MoT Overall |
|---|---|---|---|---|---|---|---|---|---|---|---|
| ❄ | ❄ | ❄ | 28.2 | 31.5 | 29.4 | 37.7 | 28.6 | 34.3 | 38.8 | 34.0 | 37.0 |
| ❄ | ❄ | 🔥 | 41.8 | 33.4 | 38.7 | 43.5 | 25.6 | 36.8 | 42.3 | 31.5 | 38.3 |
| ❄ | 🔥 | ❄ | 35.5 | 32.9 | 34.5 | 41.4 | 27.6 | 36.2 | 43.2 | 35.6 | 40.4 |
| ❄ | 🔥 | 🔥 | 45.8 | 32.9 | 41.0 | 46.8 | 25.9 | 39.0 | 46.3 | 29.8 | 40.1 |
| 🔥 | 🔥 | ❄ | 39.9 | **34.3** | 37.8 | 43.0 | **31.3** | 38.6 | 45.8 | **36.3** | **42.2** |
| 🔥 | 🔥 | 🔥 | **47.7** | 33.1 | **42.2** | **47.7** | 27.0 | **39.9** | **47.8** | 31.6 | 41.7 |

Table 2 summarizes the effects of unfreezing the image tokenizer and/or LLM during both pretraining (Stage 1) and fine-tuning (Stage 2) across different fusion architectures. Unfreezing the image tokenizer consistently provides moderate improvements in in-domain accuracy, while unfreezing the LLM backbone yields even larger gains, most notably for Joint-Decoder models, which have less capacity for multimodal integration than Cross-Attention or MoT. When both the image tokenizer and language layers are frozen, MoT outperforms the other architectures, thanks to the extra trainable parameters it has (over 400M more parameters from the image modality-transformer). However, as

more layers are unfrozen, all architectures perform more similarly. In contrast, unfreezing the LLM often leads to a reduction in out-of-domain performance, whereas unfreezing only the image tokenizer can provide a modest out-of-domain boost. The full de-aggregated results for all architectures and tokenizers can be found in Appendix A.

These results show that the optimal freezing strategy is heavily dependent on the expected downstream usage. For downstream usage that is very similar to the visual training data, unfreezing as many layers as possible, especially at the fine-tuning stage, can often bring the best results. However, for generalization to new formats and out-of-distribution data, it is helpful to leverage the knowledge in the language model gained from training on broader text data and keep the language layers frozen. In this case, we find that the Mixture-of-Transformers (MoT) architecture is particularly effective. By providing dedicated parameters for multimodal integration without increasing the FLOPs per forward pass (Liang et al., 2025; Shi et al., 2025b), MoT enables a strategy where the language model can be frozen without severely harming in-domain performance. This approach yields strong performance on both in-domain and out-of-domain tasks, preserves the LLM's original text-only capabilities, and remains computationally efficient. This finding aligns with recent work (Dai et al., 2024; Lin et al., 2024; Shi et al., 2025b) and highlights the value of strategies that limit LLM supervision, especially when high-quality training data is scarce.

**Takeaway 2.** Architectures that explicitly separate modalities such as the MoT fusion architecture, along with training recipes that preserve the more general textual knowledge and reasoning of the base language model, generalize well to out-of-domain tasks.

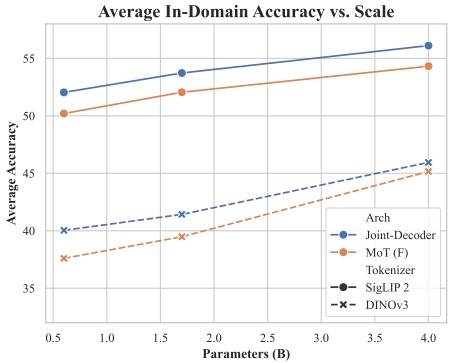 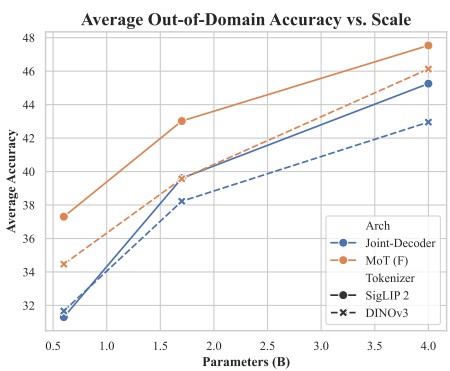

(a) In-Domain Average Performance      (b) Out-of-Domain Average Performance

**Figure 2:** **Average accuracy for in-domain and out-of-domain tasks as model scales.** These plot tracks how in-domain and out-of-domain performance changes as we scale the base LLM from 0.6B to 4B parameters. We compare the completely unfrozen Joint-Decoder architecture to the MoT architecture with frozen language layers, denoted MoT (F). We test each configuration with both SigLIP 2 and DINOv3.

To see if our takeaways still hold at larger scale, we run experiments with Qwen3-1.7B and Qwen3-4B as the LLM backbone, using either COYO-700M (Byeon et al., 2022) or DataComp-1B (Gadre et al., 2023) in Stage 1 training to account for the increased number of parameters. More details on the training setup can be found in Appendix D.

Figure 2 shows how the average accuracy changes as we scale the model size. We see that substantially higher in-domain performance for SigLIP 2 over DINOv3, regardless of scale and architecture, and we see higher out-of-domain scores as well, supporting the crucial role of the tokenizer objective in downstream performance. Between the unfrozen Joint-Decoder architecture and the MoT architecture with frozen language layers, we see that both setups scale with the size of the LLM, with the MoT architecture performing the best for out-of-domain tasks.

## 4 CROSS-MODALITY TRANSFER LEARNING

In this section, we systematically evaluate *cross-modality transfer*–when training on one modality improves performance on another modality–across the models we trained. Motivated by similar settings in prior work (Wang et al., 2024; Yamada et al., 2024; Rahmanzadehgervi et al., 2024), we construct three synthetic datasets designed to isolate reasoning from perception. Each dataset

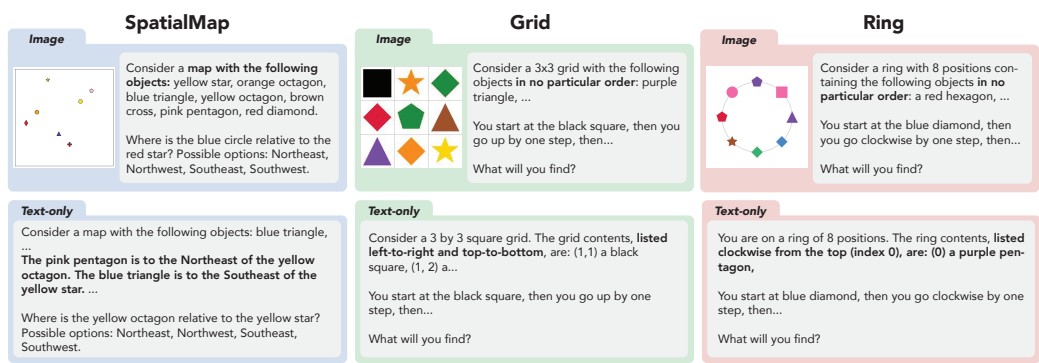

**Figure 3: Examples from the three synthetic datasets used to study cross-modality transfer.** Each dataset provides paired image and text-only versions with equivalent information but distinct spatial structures. **SpatialMap** (left, blue) places objects on an open canvas and asks about relative positions between two objects. **Grid** (middle, green) arranges objects in a 3×3 lattice and requires tracking movement across cells. **Ring** (right, red) positions objects cyclically and requires tracking clockwise or counterclockwise traversal.

pairs procedurally generated images with equivalent text-only representations and is built around a distinct spatial reasoning task: SpatialMap (objects on an open canvas), Grid (objects in a 2D grid), and Ring (objects arranged cyclically). See Figure 3 for examples from each dataset. Because the image-based (VQA) and text-only (textual QA, or TQA) versions of each dataset contain identical information, we can directly measure a model's ability to transfer learned concepts across modalities. The datasets in Wang et al. (2024) are purely for benchmarking and only contain test splits, but we wanted to evaluate how architecture choices and design decisions impact cross-modality transfer when fine-tuning on open-weight models. As a result, we generated our own version of the datasets from Wang et al. (2024) with train and test splits. Each of our image-based and text-only datasets provides 4,500 training examples, 500 in-distribution (**InD**) test examples, and 500 out-of-distribution (**OOD**) test examples that are designed to be compositionally harder (e.g. having more objects or longer navigation sequences). Further details on dataset generation are in Appendix C.

To measure transfer, we perform further fine-tuning (as described in Section 2.3) on either the image-based (VQA) or text-only (TQA) version of a task. We then evaluate each fine-tuned model across two axes of generalization:

- **Same-Modality vs. Cross-Modality:** We test on tasks using the same modality as the synthetic data (e.g., VQA→VQA) and on tasks using the other modality (e.g., VQA→TQA).
- **In-Distribution (InD) vs. Out-of-Distribution (OOD):** We test on both the standard test set (InD) and the more challenging, compositionally distinct test set (OOD).

We analyze image to text cross-modality transfer (VQA→TQA) in Section 4.1 and text to image cross-modality transfer (TQA→VQA) in Section 4.2. In addition, an analogous analysis performed on several open-weight VLMs is provided in Appendix B.

### 4.1 CROSS-MODALITY TRANSFER FROM IMAGE TO TEXT

To evaluate image-to-text cross-modality transfer, we fine-tune our stage-2 models separately on the image-based (VQA) and text-only (TQA) versions of our synthetic datasets (SpatialMap, Grid, and Ring). We then evaluate performance of both VQA-finetuned and TQA-finetuned models on the TQA version of the task. This allows us to measure how well knowledge acquired from visual inputs transfers to a purely textual domain (VQA→TQA) and compare it with when it is finetuned on the same-modality task (TQA→TQA). The distinction between image-based and text-only tasks is illustrated in Figure 3.

Figure 4 presents these results, averaged across all fusion architectures to isolate the impact of the image tokenizer. Despite training on TQA, the tokenizers generally improve from their untrained baselines and perform better than random even on OOD tasks, showing transfer from the visual domain.

We observe that the degree of transfer is highly dependent on the representational alignment between the image and text modalities of a given dataset. For the Grid and Ring datasets, where object

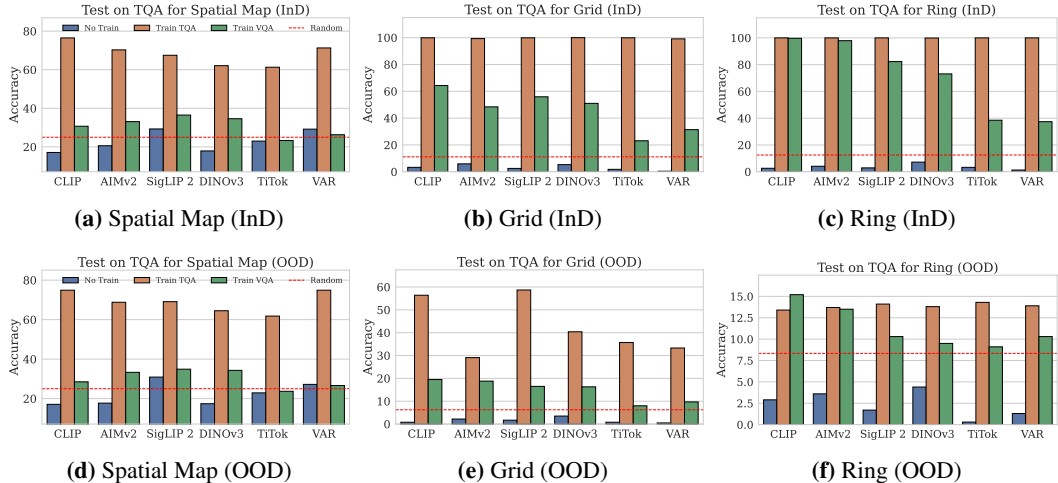

**Figure 4: Image-to-Text (VQA→TQA) transfer performance across image tokenizers.** Scores are averaged across all three fusion architectures. Despite training on a different modality, the models generally improve from their untrained baselines and perform better than random, indicating visual transfer. Transfer is better for datasets with strong representational alignment like Grid and Ring, where we see that training with a different modality can sometimes perform as well as training with the same modality.

locations are heavily structured and the textual description is a direct, one-to-one serialization of the visual information, models achieve strong cross-modality transfer. This is especially true for the Ring dataset, where cross-modality training sometimes performed as well as same-modality training on both the in-distribution and out-of-distribution tests and even achieved near-perfect accuracy in a few cases. For the SpatialMap dataset, where the text describes relative spatial locations, the textual description of relative coordinates is ambiguous and does not uniquely define the visual layout. Subsequently, transfer is significantly more challenging, resulting in text-only accuracy not substantially higher than random, especially when compared to the other two datasets. This suggests that the representational alignment between modalities is a key bottleneck.

One can find the full, de-aggregated VQA→TQA transfer results for each architecture and tokenizer in Table 6 in Section A.2.

### 4.2 Cross-Modality Transfer from Text to Image

To evaluate text-to-image transfer, we fine-tune our stage-2 models separately on the text-only (TQA) and image-based (VQA) versions of our synthetic datasets and then evaluate performance of each models on the VQA tasks. This enables us to measure how well knowledge acquired from textual inputs translates to visual understanding (TQA→VQA), which we can then compare with same-modality training (VQA→VQA).

For both in-distribution and out-of-distribution tasks, the results, grouped by image tokenizer in Figure 5, reveal a stark asymmetry compared to the previous subsection. The models are able to learn each visual task with same-modality training, generally performing better than random and often attaining near-perfect accuracy. However, we do not see this performance when transferring textual knowledge to the image domain through cross-modality training, where performance tended to be worse than random guessing. This is expected with the current modular practice of building VLMs as text-only fine-tuning provides no gradient signal to update the image tokenizer or its alignment with the language model. This highlights an fundamental limitation of current VLM architectures as updating existing multimodal alignment necessitates multimodal data. Addressing this remains an avenue for future investigation. One can find the full, de-aggregated TQA→VQA transfer results for each architecture and tokenizer in Table 7 in Section A.2.

Overall, our cross-modality experiments yield the following conclusion:

**Takeaway 3.** Cross-modality transfer is heavily dependent on representational alignment between the text and images. Image-to-text (VQA→TQA) transfer on our synthetic tasks is comparatively strong, whereas we were unable to observe text-to-image (TQA→VQA) transfer.

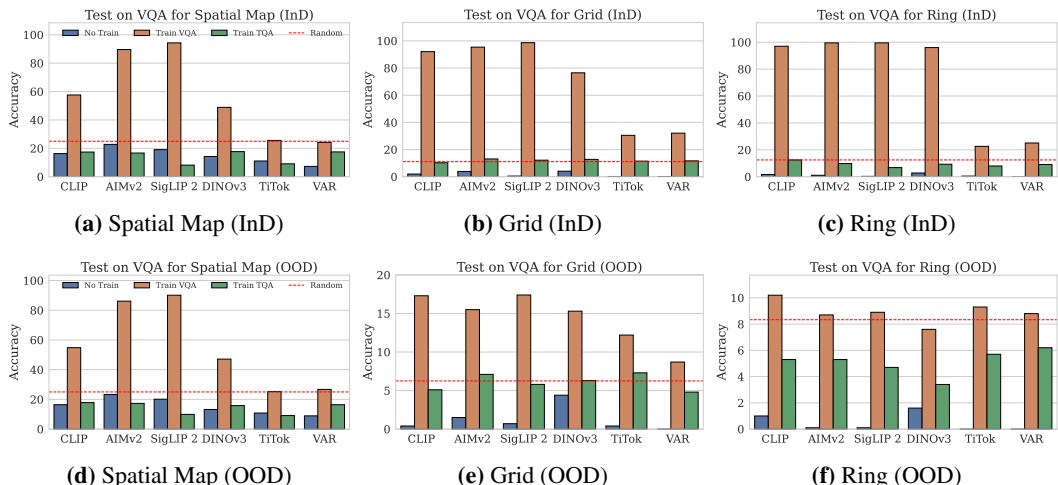

**Figure 5: Text-to-Image (TQA→VQA) transfer performance across image tokenizers.** Scores are averaged across all three fusion architectures. Despite the tasks being learnable, as evidenced by the same modality training achieving better than random scores, we find that different modality training gives poor results, usually performing worse than random, indicating a lack of transfer.

## 5 RELATED WORK

Our work builds on three key areas of VLM research: the design of image tokenizers, the diversity of training recipes, and direct architectural comparisons. At a higher level, a rich body of survey work has mapped out the design space of pretraining from scratch models with joint vision-language representation and training VLMs from existing pretrained image tokenizers and LLMs. Early surveys (Long et al., 2022; Du et al., 2022) focus on joint vision–language pretraining pipelines, covering data encoding strategies, backbone choices, fusion architectures, and pretraining objectives. More recent overviews (Yin et al., 2024; Zhang et al., 2024) center on VLMs (also referred to as "multimodal LLMs") in which they detail practices on building off of strong language backbones and catalog emerging adapter designs, alignment strategies, and evaluation protocols. Meanwhile, Ma et al. (2025) provides a complementary perspective organized around training paradigms for efficiently integrating visual perception into LLMs. Different from the survey papers that mainly relay results within the literature under non-standardized training settings, our work offers a controlled and rigorous empirical study that isolates how tokenizers, fusion architectures, and layer-freezing strategies interact.

**The Importance of the Image Tokenizer.** The choice of image tokenizer is a critical factor in VLM performance. Prior work has explored this from several angles. For instance, PaLI (Chen et al., 2023) demonstrated the benefits of scaling up the vision encoder, while Eagle (Shi et al., 2025a) compared the effect of tokenizer objectives and freezing in Joint-Decoders and improved performance by combining multiple task-specific image tokenizers. More recent studies, such as Cambrian-1 (Tong et al., 2024), have focused on comparing different language-supervised tokenizers. While these works establish the tokenizer's importance, a systematic comparison of tokenizers trained with different objectives (e.g., contrastive vs. reconstructive) across varied architectural and training setups remains an open area. Our work addresses this by evaluating six distinct tokenizers across three fusion architectures.

**Diversity in Training Recipes.** Training recipes for VLMs are highly varied, typically involving multi-stage protocols with different layer-freezing strategies. A common approach, seen in the LLaVA family (Liu et al., 2023; 2024; Li et al., 2025), InternVL 2.5 (Chen et al., 2024), and DeepSeek-VL (Lu et al., 2024a), involves freezing both the image tokenizer and LLM during an initial alignment stage where only a small adapter is trained. In contrast, models like Qwen 2.5 VL (Bai et al., 2025) and DeepSeek-VL2 (Wu et al., 2024) unfreeze the image tokenizer from the start to foster better alignment, while keeping the LLM frozen. A less common end-to-end approach, adopted by Molmo (Deitke et al., 2024), trains all parameters simultaneously but requires high-quality data and carefully tuned learning rates. The choices become even more complex in later fine-tuning stages, as shown by Cambrian-1, which ablates freezing the image encoder during

its second training phase and finds it can be beneficial. This diversity makes it difficult to attribute performance gains to specific training choices versus other confounding factors. Our work addresses this by systematically ablating freezing strategies across fixed architectures.

**Architectural Comparisons.** Direct architectural comparisons for VLMs have been conducted, but often with limited scope. For example, NVLM (Dai et al., 2024) provided an early comparison between the Joint-Decoder and Cross-Attention architectures, along with ablations on using special tokens for visual knowledge transfer. More recently, LMFusion (Shi et al., 2025b) compared the Joint-Decoder against using modality-specific MLPs or an MoT architecture, but their study was confined to a setting where the language backbone remained frozen. Some survey papers highlight the proliferation of such adapter-based and fusion architectures, while also noting the scarcity of tightly controlled head-to-head comparisons across design choices (Yin et al., 2024; Zhang et al., 2024). Our work expands on these studies by providing a unified comparison of all three major architectures (Joint-Decoder, Cross-Attention, and MoT) while also varying layer-freezing strategies for both the vision and language components, thereby offering a more comprehensive understanding of architectural trade-offs.

Finally, our cross-modality transfer analysis connects to recent work showing that VLMs often fail to faithfully verbalize fine-grained visual information even when their vision encoders contain the relevant signals (Rahmanzadehgervi et al., 2024; Fu et al., 2025). Our controlled comparisons across tokenizers and fusion architectures reveal how this gap widens or narrows as we change the multi-modal integration mechanism and training recipe.

## 6 CONCLUDING REMARKS

In this work, we systematically studied how the choice of image tokenizer, fusion architecture, and layer freezing strategies influence the downstream performance of vision-language models. Our results highlight the critical impact of the image tokenizer choice, reveal distinct in-domain and out-of-domain trade-offs associated with layer freezing, quantify the robustness of different fusion architectures to various freezing strategies, and demonstrate the varying degrees of cross-domain transfer enabled by these design choices. Taken together, our ablations point to three overarching design principles: first, image tokenizers trained with text-aware objectives are crucial for strong VLM performance, outperforming those trained without such objectives on both in-domain and out-of-domain tasks; second, architectures that explicitly separate modalities such as the Mixture-of-Transformers fusion architecture, along with training recipes that preserve the more general textual knowledge and reasoning of the base language model, generalize well to out-of-domain tasks; finally, cross-modality transfer is generally weak unless the image and text representations are tightly aligned and structured, with image-to-text transfer on synthetic reasoning tasks proving more reliable than the reverse.

Several open questions arise from our findings. Our experiments were primarily conducted with models in the 1–1.5B parameter range, and we find that our observations hold for LLM backbones with up to 4B parameters. It is important to explore how these results scale to even larger models. A key direction is to better understand *why* full LLM fine-tuning often harms out-of-distribution robustness—for example, whether it induces catastrophic forgetting of visual grounding, encourages shortcut text patterns that overfit to the fine-tuning distribution, or misaligns the latent spaces of the underlying vision components. A better understanding of the vision latents produced by various tokenizer objectives and their use in downstream language modeling would also be helpful for creating stronger tokenizers for VLMs and developing end-to-end VLM training strategies. Future work could also develop training recipes and objectives that more explicitly encourage cross-modality alignment, such as auxiliary cross-modal consistency losses, curriculum strategies that interleave modalities, or architectures that expose shared symbolic or structured intermediate representations.

Beyond vision and text, future work could extend the comparison to other modalities, such as speech, to investigate whether transfer occurs more readily between specific modality pairs or domains (e.g., speech and poetry vs. vision and physical reasoning), and to test whether our observations about MoT-style fusion and partial unfreezing generalize. Additionally, our models are text-only generators; further research is needed to assess the trade-offs of different architectures and training recipes for other generative tasks, and to determine whether the same design principles continue to hold in more complex multimodal generation settings.

ETHICS STATEMENT

This work does not raise any ethical concerns.

REPRODUCIBILITY STATEMENT

To facilitate reproducibility and future research, we make the following resources available. Our full codebase, all trained model checkpoints, and our newly introduced synthetic datasets will be made publicly available upon publication. Detailed hyperparameters for all training stages are provided in Appendix D, and the generation procedure for our synthetic datasets is described in Appendix C. All academic benchmarks and the Qwen3-0.6B base model used in this work are already publicly available.

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

# A  FULL EVALUATIONS

We include in this appendix the full list of evaluations for all of the models that we trained for stage 2 and stage 3.

## A.1  STAGE 2 EVALUATIONS: ACADEMIC BENCHMARKS

**Table 3: Detailed evaluation results (accuracy in %) for the Joint-Decoder models.** The table presents scores on individual benchmarks alongside in-domain, out-of-domain, and overall averages. We ablate the choice of image tokenizer and whether the image tokenizer (**Im**) or language layers (**La**) are frozen (❄️) or unfrozen (🔥) over the two stages (**S1** and **S2**). The highest score in each column is bolded.

| Tokenizer | S1 Im | S2 Im | S2 La | VQAv2 test-dev | A-OKVQA val | ChartQA test | TextVQA val | DocVQA test | Average In-domain | MathVista testmini | MMTBench val | RealWorldQA | Average Out-of-domain | Average All |
|---|---|---|---|---|---|---|---|---|---|---|---|---|---|---|
| CLIP | ❄️ | ❄️ | ❄️ | 49.8 | 22.1 | 12.2 | 20.1 | 12.1 | 23.2 | 25.0 | 37.4 | 29.0 | 30.5 | 25.9 |
|  | ❄️ | ❄️ | 🔥 | 66.7 | 41.6 | 20.8 | 31.9 | 20.7 | 36.3 | 27.3 | 34.3 | 37.0 | 32.9 | 35.0 |
|  | ❄️ | 🔥 | ❄️ | 61.8 | 31.4 | 20.4 | 27.9 | 16.5 | 31.6 | 26.7 | 34.7 | 34.5 | 32.0 | 31.7 |
|  | ❄️ | 🔥 | 🔥 | 72.1 | 45.4 | 29.2 | 35.9 | 24.8 | 41.5 | 27.7 | 35.0 | 35.9 | 32.9 | 38.3 |
|  | 🔥 | 🔥 | ❄️ | 66.0 | 35.7 | 23.0 | 34.1 | 20.4 | 35.8 | 22.8 | 38.5 | 31.4 | 30.9 | 34.0 |
|  | 🔥 | 🔥 | 🔥 | 74.1 | 47.2 | 30.6 | 40.1 | 25.2 | 43.4 | 28.4 | 35.4 | 36.9 | 33.6 | 39.7 |
| AIMv2 | ❄️ | ❄️ | ❄️ | 62.5 | 30.3 | 19.5 | 33.6 | 17.9 | 32.8 | 26.5 | **43.7** | 26.9 | 32.4 | 32.6 |
|  | ❄️ | ❄️ | 🔥 | 75.3 | 49.5 | 30.6 | 43.2 | 25.2 | 44.7 | 28.2 | 37.0 | 38.3 | 34.5 | 40.9 |
|  | ❄️ | 🔥 | ❄️ | 68.0 | 35.5 | 28.1 | 37.1 | 21.8 | 38.1 | 26.9 | 41.9 | 37.1 | 35.3 | 37.1 |
|  | ❄️ | 🔥 | 🔥 | 77.0 | 50.0 | 36.3 | 43.9 | 27.4 | 46.9 | 26.2 | 36.1 | 41.0 | 34.4 | 42.2 |
|  | 🔥 | 🔥 | ❄️ | 69.1 | 38.1 | 29.6 | 38.8 | 22.6 | 39.6 | 26.4 | 43.5 | **42.7** | **37.6** | 38.9 |
|  | 🔥 | 🔥 | 🔥 | 77.0 | 50.4 | 37.2 | 45.5 | 28.4 | 47.7 | **29.2** | 36.7 | 37.4 | 34.4 | 42.7 |
| SigLIP 2 | ❄️ | ❄️ | ❄️ | 55.9 | 24.0 | 15.7 | 30.0 | 17.0 | 28.5 | 25.5 | 38.1 | 31.4 | 31.7 | 29.7 |
|  | ❄️ | ❄️ | 🔥 | 74.8 | 47.2 | 28.8 | 43.6 | 26.6 | 44.2 | 28.0 | 38.1 | 32.4 | 32.8 | 39.9 |
|  | ❄️ | 🔥 | ❄️ | 65.4 | 31.4 | 29.3 | 35.5 | 22.2 | 36.8 | 24.1 | 37.7 | 32.7 | 31.5 | 34.8 |
|  | ❄️ | 🔥 | 🔥 | 77.8 | 50.2 | 43.2 | 44.7 | 29.6 | 49.1 | 25.1 | 35.7 | 33.2 | 31.3 | 42.4 |
|  | 🔥 | 🔥 | ❄️ | 71.1 | 40.9 | 36.5 | 45.1 | 28.1 | 44.3 | 27.4 | 42.1 | 34.1 | 34.5 | 40.7 |
|  | 🔥 | 🔥 | 🔥 | **78.4** | **52.3** | **46.8** | **50.0** | **32.7** | **52.0** | 28.6 | 35.6 | 29.5 | 31.3 | **44.2** |
| DINOv3 | ❄️ | ❄️ | ❄️ | 53.6 | 22.8 | 9.6 | 12.7 | 9.6 | 21.7 | 24.8 | 37.3 | 28.5 | 30.2 | 24.9 |
|  | ❄️ | ❄️ | 🔥 | 71.5 | 45.7 | 19.0 | 20.5 | 15.8 | 34.5 | 25.5 | 31.7 | 31.9 | 29.7 | 32.7 |
|  | ❄️ | 🔥 | ❄️ | 61.2 | 28.5 | 15.0 | 16.9 | 10.8 | 26.5 | 24.7 | 37.7 | 38.7 | 33.7 | 29.2 |
|  | ❄️ | 🔥 | 🔥 | 74.7 | 45.8 | 26.0 | 23.1 | 17.4 | 37.4 | 27.8 | 33.0 | 30.6 | 30.5 | 34.8 |
|  | 🔥 | 🔥 | ❄️ | 63.8 | 30.0 | 17.5 | 22.8 | 13.1 | 29.4 | 24.2 | 39.1 | 31.4 | 31.6 | 30.2 |
|  | 🔥 | 🔥 | 🔥 | 75.6 | 47.2 | 28.7 | 29.1 | 19.6 | 40.0 | 30.3 | 33.5 | 31.2 | 31.7 | 36.9 |
| TiTok | ❄️ | ❄️ | ❄️ | 3.2 | 0.2 | 5.1 | 1.0 | 3.0 | 2.5 | 23.8 | 27.7 | 28.6 | 26.7 | 11.6 |
|  | ❄️ | ❄️ | 🔥 | 43.1 | 26.0 | 13.9 | 11.8 | 11.8 | 21.3 | 26.0 | 22.8 | 32.5 | 27.1 | 23.5 |
| VAR | ❄️ | ❄️ | ❄️ | 30.3 | 2.0 | 9.2 | 4.9 | 6.0 | 10.5 | 24.4 | 27.5 | 32.3 | 28.1 | 17.1 |
|  | ❄️ | ❄️ | 🔥 | 46.5 | 27.7 | 13.7 | 11.9 | 11.9 | 22.3 | 26.9 | 19.3 | 21.0 | 22.4 | 22.4 |

Table 4: **Detailed evaluation results (accuracy in %) for the Cross-Attention models.** The table presents scores on individual benchmarks alongside in-domain, out-of-domain, and overall averages. We ablate the choice of image tokenizer and whether the image tokenizer (**Im**) or language layers (**La**) are frozen (❄) or unfrozen (🔥) over the two stages (**S1** and **S2**). The highest score in each column is bolded.

| Tokenizer | S1 Im | S2 Im | S2 La | VQAv2 test-dev | A-OKVQA val | ChartQA test | TextVQA val | DocVQA test | Average In-domain | MathVista testmini | MMTBench val | RealWorldQA | Average Out-of-domain | Average All |
|---|---|---|---|---|---|---|---|---|---|---|---|---|---|---|
| CLIP | ❄ | ❄ | ❄ | 64.4 | 34.5 | 19.7 | 33.6 | 19.7 | 34.4 | 24.7 | 28.4 | 36.6 | 29.9 | 32.7 |
| | ❄ | ❄ | 🔥 | 72.5 | 46.0 | 24.0 | 37.1 | 23.0 | 40.5 | 24.7 | 19.3 | 30.3 | 24.8 | 34.6 |
| | ❄ | 🔥 | ❄ | 67.3 | 37.2 | 27.4 | 33.9 | 22.0 | 37.6 | 25.0 | 31.1 | 32.9 | 29.7 | 34.6 |
| | ❄ | 🔥 | 🔥 | 74.8 | 45.8 | 33.4 | 39.7 | 25.9 | 43.9 | 25.8 | 23.4 | 37.3 | 28.8 | 38.3 |
| | 🔥 | 🔥 | ❄ | 69.7 | 40.4 | 30.6 | 39.5 | 24.2 | 40.9 | 26.5 | 28.1 | 33.7 | 29.5 | 36.6 |
| | 🔥 | 🔥 | 🔥 | 74.5 | 47.5 | 33.4 | 42.7 | 26.9 | 45.0 | 27.5 | 20.6 | 30.7 | 26.3 | 38.0 |
| AIMv2 | ❄ | ❄ | ❄ | 69.4 | 36.7 | 24.8 | 40.1 | 21.1 | 38.4 | 24.0 | 26.9 | 28.9 | 26.6 | 34.0 |
| | ❄ | ❄ | 🔥 | 75.4 | 47.0 | 30.7 | 43.6 | 25.3 | 44.4 | 26.9 | 29.7 | 25.0 | 27.2 | 37.9 |
| | ❄ | 🔥 | ❄ | 71.1 | 40.6 | 30.6 | 40.0 | 22.2 | 40.9 | 24.4 | 26.7 | 34.4 | 28.5 | 36.3 |
| | ❄ | 🔥 | 🔥 | 76.7 | 49.8 | 35.9 | 43.5 | 25.8 | 46.4 | 27.2 | 28.9 | 28.1 | 28.1 | 39.5 |
| | 🔥 | 🔥 | ❄ | 72.0 | 40.5 | 30.8 | 41.8 | 23.6 | 41.7 | 25.0 | 32.5 | 34.8 | 30.7 | 37.6 |
| | 🔥 | 🔥 | 🔥 | 76.5 | 49.8 | 36.1 | 45.3 | 26.9 | 46.9 | 26.1 | 20.4 | 25.0 | 23.8 | 38.3 |
| SigLIP 2 | ❄ | ❄ | ❄ | 70.4 | 37.8 | 23.3 | 44.8 | 25.2 | 40.3 | 24.5 | 33.0 | 30.6 | 29.4 | 36.2 |
| | ❄ | ❄ | 🔥 | 76.5 | 49.1 | 27.2 | 47.4 | 27.7 | 45.6 | 26.2 | 20.8 | 27.3 | 24.8 | 37.8 |
| | ❄ | 🔥 | ❄ | 72.8 | 42.9 | 38.8 | 45.0 | 28.7 | 45.7 | 26.1 | 24.9 | 22.6 | 24.6 | 37.7 |
| | ❄ | 🔥 | 🔥 | 78.1 | **50.6** | 43.0 | 48.1 | 30.7 | 50.1 | 25.3 | 17.2 | 19.6 | 20.7 | 39.1 |
| | 🔥 | 🔥 | ❄ | 73.5 | 43.2 | 39.2 | 47.4 | 29.4 | 46.5 | 25.9 | **36.1** | **39.1** | **33.7** | 41.7 |
| | 🔥 | 🔥 | 🔥 | **78.2** | 49.8 | **44.2** | **50.7** | **32.6** | **51.1** | **29.2** | 30.3 | 33.5 | 31.0 | **43.6** |
| DINOv3 | ❄ | ❄ | ❄ | 60.6 | 29.1 | 12.0 | 16.1 | 11.5 | 25.9 | 25.6 | 24.1 | 28.0 | 25.9 | 25.9 |
| | ❄ | ❄ | 🔥 | 70.5 | 42.9 | 17.1 | 19.9 | 16.2 | 33.3 | 25.0 | 17.2 | 23.1 | 21.8 | 29.0 |
| | ❄ | 🔥 | ❄ | 65.3 | 33.8 | 18.0 | 18.4 | 12.6 | 29.6 | 24.1 | 26.6 | 24.1 | 24.9 | 27.9 |
| | ❄ | 🔥 | 🔥 | 73.2 | 44.9 | 21.6 | 22.1 | 17.6 | 35.9 | 24.6 | 16.4 | 22.6 | 21.2 | 30.4 |
| | 🔥 | 🔥 | ❄ | 68.3 | 36.2 | 21.2 | 26.4 | 15.7 | 33.6 | 24.9 | 32.4 | 31.9 | 29.7 | 32.1 |
| | 🔥 | 🔥 | 🔥 | 74.6 | 46.8 | 26.4 | 29.5 | 19.7 | 39.4 | 23.9 | 20.6 | 35.6 | 26.7 | 34.6 |
| TiTok | ❄ | ❄ | ❄ | 40.0 | 13.9 | 10.6 | 7.7 | 8.2 | 16.1 | 24.7 | 25.0 | 37.5 | 29.1 | 20.9 |
| | ❄ | ❄ | 🔥 | 42.5 | 24.6 | 13.6 | 11.1 | 11.8 | 20.7 | 23.8 | 21.3 | 25.8 | 23.6 | 21.8 |
| VAR | ❄ | ❄ | ❄ | 42.4 | 16.3 | 10.6 | 8.5 | 9.1 | 17.4 | 23.7 | 24.4 | 36.9 | 28.3 | 21.5 |
| | ❄ | ❄ | 🔥 | 46.6 | 24.6 | 13.2 | 11.6 | 11.9 | 21.6 | 26.3 | 22.5 | 24.8 | 24.6 | 22.7 |

**Table 5: Detailed evaluation results (accuracy in %) for the Mixture-of-Transformers models.** The table presents scores on individual benchmarks alongside in-domain, out-of-domain, and overall averages. We ablate the choice of image tokenizer and whether the image tokenizer (**Im**) or language layers (**La**) are frozen ( ❄️ ) or unfrozen ( 🔥 ) over the two stages (**S1** and **S2**). The highest score in each column is bolded.

| Tokenizer | S1 Im | S2 Im | S2 La | VQAv2 test-dev | A-OKVQA val | ChartQA test | TextVQA val | DocVQA test | Average In-domain | MathVista testmini | MMTBench val | RealWorldQA | Average Out-of-domain | Average All |
|---|---|---|---|---|---|---|---|---|---|---|---|---|---|---|
| CLIP | ❄️ | ❄️ | ❄️ | 63.1 | 35.7 | 20.3 | 30.5 | 18.9 | 33.7 | 27.2 | 38.2 | 33.6 | 33.0 | 33.4 |
|  | ❄️ | ❄️ | 🔥 | 67.0 | 42.8 | 22.3 | 33.9 | 21.5 | 37.5 | 28.4 | 31.0 | 22.1 | 27.1 | 33.6 |
|  | ❄️ | 🔥 | ❄️ | 69.5 | 40.1 | 26.6 | 34.0 | 23.5 | 38.7 | 27.5 | 40.6 | 40.8 | 36.3 | 37.8 |
|  | ❄️ | 🔥 | 🔥 | 72.6 | 46.7 | 29.8 | 36.7 | 24.3 | 42.0 | 27.0 | 31.4 | 29.0 | 29.1 | 37.2 |
|  | 🔥 | 🔥 | ❄️ | 72.1 | 46.4 | 29.1 | 37.1 | 25.1 | 42.0 | 26.1 | 41.0 | 43.4 | 36.8 | 40.0 |
|  | 🔥 | 🔥 | 🔥 | 74.2 | 47.4 | 30.4 | 39.5 | 25.7 | 43.4 | 28.2 | 34.6 | 33.9 | 32.2 | 39.2 |
| AIMv2 | ❄️ | ❄️ | ❄️ | 72.8 | 42.5 | 28.5 | 40.4 | 23.8 | 41.6 | 27.8 | 40.4 | 36.9 | 35.0 | 39.1 |
|  | ❄️ | ❄️ | 🔥 | 75.4 | 48.0 | 30.7 | 43.4 | 26.4 | 44.8 | 27.3 | 40.8 | 42.2 | 36.8 | 41.8 |
|  | ❄️ | 🔥 | ❄️ | 74.7 | 44.9 | 35.2 | 41.4 | 26.8 | 44.6 | 29.0 | **41.9** | **42.4** | **37.8** | 42.0 |
|  | ❄️ | 🔥 | 🔥 | 77.3 | 50.3 | 37.0 | 44.1 | 28.0 | 47.3 | 27.1 | 31.0 | 28.0 | 28.7 | 40.3 |
|  | 🔥 | 🔥 | ❄️ | 74.8 | 46.6 | 35.1 | 42.5 | 27.6 | 45.3 | 26.4 | 39.0 | 39.0 | 34.8 | 41.4 |
|  | 🔥 | 🔥 | 🔥 | 77.1 | 51.1 | 36.9 | 44.3 | 28.2 | 47.5 | 26.4 | 32.2 | 32.9 | 30.5 | 41.2 |
| SigLIP 2 | ❄️ | ❄️ | ❄️ | 71.3 | 42.9 | 26.6 | 41.2 | 23.9 | 41.2 | 25.8 | 38.6 | 37.3 | 33.9 | 38.4 |
|  | ❄️ | ❄️ | 🔥 | 75.0 | 48.1 | 28.8 | 43.7 | 27.4 | 44.6 | 27.5 | 34.3 | 29.5 | 30.4 | 39.3 |
|  | ❄️ | 🔥 | ❄️ | 74.7 | 43.9 | 40.6 | 43.0 | 28.8 | 46.2 | 26.7 | 39.3 | 31.9 | 32.6 | 41.1 |
|  | ❄️ | 🔥 | 🔥 | 78.0 | 49.6 | 43.6 | 45.4 | 30.4 | 49.4 | 28.2 | 34.6 | 32.0 | 31.6 | 42.7 |
|  | 🔥 | 🔥 | ❄️ | 76.4 | 47.9 | 43.9 | 49.1 | **33.7** | 50.2 | 28.4 | 41.6 | 42.0 | 37.3 | **45.4** |
|  | 🔥 | 🔥 | 🔥 | **79.0** | **52.7** | **46.6** | **50.7** | 33.1 | **52.4** | **29.4** | 36.2 | 30.7 | 32.1 | 44.8 |
| DINOv3 | ❄️ | ❄️ | ❄️ | 68.2 | 40.4 | 17.9 | 18.6 | 14.8 | 32.0 | 25.1 | 38.8 | 36.9 | 33.6 | 32.6 |
|  | ❄️ | ❄️ | 🔥 | 71.8 | 44.0 | 18.0 | 20.2 | 16.6 | 34.1 | 23.9 | 20.3 | 21.7 | 22.0 | 29.6 |
|  | ❄️ | 🔥 | ❄️ | 71.5 | 40.3 | 23.1 | 21.4 | 15.6 | 34.4 | 23.8 | 39.9 | 37.9 | 33.9 | 34.2 |
|  | ❄️ | 🔥 | 🔥 | 75.4 | 47.6 | 27.1 | 23.2 | 17.1 | 38.1 | 26.9 | 33.4 | 27.7 | 29.3 | 34.8 |
|  | 🔥 | 🔥 | ❄️ | 73.2 | 42.4 | 25.4 | 27.8 | 19.2 | 37.6 | 25.8 | 40.6 | 37.0 | 34.5 | 36.4 |
|  | 🔥 | 🔥 | 🔥 | 75.9 | 47.8 | 29.4 | 28.8 | 19.4 | 40.2 | 26.1 | 33.4 | 35.8 | 31.8 | 37.1 |
| TiTok | ❄️ | ❄️ | ❄️ | 39.7 | 21.0 | 10.5 | 7.7 | 7.9 | 17.4 | 23.6 | 27.8 | 30.3 | 27.2 | 21.1 |
|  | ❄️ | ❄️ | 🔥 | 42.3 | 26.9 | 12.9 | 10.5 | 11.2 | 20.8 | 24.9 | 20.4 | 19.3 | 21.5 | 21.1 |
| VAR | ❄️ | ❄️ | ❄️ | 41.9 | 23.8 | 10.8 | 8.7 | 9.6 | 19.0 | 25.2 | 27.1 | 32.0 | 28.1 | 22.4 |
|  | ❄️ | ❄️ | 🔥 | 46.9 | 29.0 | 13.4 | 11.7 | 12.4 | 22.7 | 25.0 | 22.4 | 24.7 | 24.0 | 23.2 |

## A.2 STAGE 3 EVALUATIONS: CROSS-MODALITY TRANSFER LEARNING

**Table 6: Image-to-Text (VQA→TQA) transfer performance (accuracy in %) by different combination of fusion architectures and image tokenizers.** Scores are evaluated after models were fine-tuned on the image-based (VQA) versions of the synthetic datasets. Each score is presented alongside the original score before the additional training.

| Dataset | Architecture | Tokenizers | VQA (InD) | | VQA (OOD) | | TQA (InD) | | TQA (OOD) | |
|---|---|---|---|---|---|---|---|---|---|---|
| | | | No Train | SFT | No Train | SFT | No Train | SFT | No Train | SFT |
| SpatialMap | Joint-Decoder | CLIP | 24.4 | 52.0 | 24.0 | 47.8 | 22.4 | 28.2 | 22.2 | 30.6 |
| | | AIMv2 | 24.8 | 88.4 | 25.6 | 85.6 | 36.4 | 37.2 | 35.0 | 37.0 |
| | | SigLIP 2 | 24.8 | 97.8 | 25.8 | 93.2 | 38.4 | 39.2 | 40.0 | 36.2 |
| | | DINOv3 | 25.6 | 59.0 | 23.4 | 57.6 | 36.4 | 33.6 | 37.6 | 33.8 |
| | | TiTok | 1.4 | 24.6 | 1.4 | 26.6 | 13.0 | 8.4 | 13.0 | 7.8 |
| | | VAR | 0.0 | 25.8 | 0.2 | 26.4 | 29.8 | 26.0 | 25.4 | 25.2 |
| | Cross Attention | CLIP | 2.6 | 88.6 | 3.0 | 83.4 | 0.2 | 30.8 | 0.0 | 25.6 |
| | | AIMv2 | 21.2 | 88.8 | 23.0 | 84.8 | 6.0 | 29.6 | 4.2 | 28.6 |
| | | SigLIP 2 | 6.8 | 88.0 | 9.0 | 83.4 | 20.6 | 37.8 | 21.2 | 36.4 |
| | | DINOv3 | 1.0 | 25.0 | 0.6 | 26.0 | 7.0 | 31.8 | 5.8 | 28.0 |
| | | TiTok | 25.2 | 28.6 | 25.8 | 24.4 | 26.8 | 35.0 | 27.8 | 39.2 |
| | | VAR | 16.8 | 22.2 | 17.2 | 28.0 | 31.6 | 28.0 | 27.8 | 28.2 |
| | MoT | CLIP | 22.0 | 32.2 | 22.2 | 33.2 | 28.8 | 33.2 | 29.0 | 29.4 |
| | | AIMv2 | 22.4 | 91.6 | 21.4 | 87.8 | 19.4 | 32.6 | 13.8 | 34.2 |
| | | SigLIP 2 | 25.6 | 97.2 | 25.6 | 93.8 | 29.0 | 32.6 | 31.4 | 32.2 |
| | | DINOv3 | 16.2 | 62.8 | 15.6 | 57.6 | 10.2 | 38.4 | 8.8 | 41.2 |
| | | TiTok | 6.6 | 23.6 | 5.2 | 24.6 | 29.2 | 26.4 | 28.0 | 24.0 |
| | | VAR | 5.2 | 24.6 | 9.2 | 25.8 | 26.2 | 24.8 | 28.4 | 26.4 |
| Ring | Joint-Decoder | CLIP | 5.0 | 96.8 | 3.0 | 9.0 | 0.4 | 100.0 | 3.4 | 14.6 |
| | | AIMv2 | 0.8 | 99.8 | 0.0 | 7.4 | 5.6 | 99.8 | 3.4 | 14.8 |
| | | SigLIP 2 | 0.8 | 99.8 | 0.4 | 6.4 | 8.6 | 97.8 | 4.8 | 9.0 |
| | | DINOv3 | 0.4 | 99.0 | 0.0 | 8.4 | 8.2 | 97.2 | 3.4 | 10.4 |
| | | TiTok | 0.6 | 20.2 | 0.0 | 8.6 | 9.0 | 13.6 | 0.0 | 7.2 |
| | | VAR | 0.0 | 24.2 | 0.0 | 8.4 | 3.0 | 44.0 | 3.8 | 11.4 |
| | Cross Attention | CLIP | 0.0 | 99.4 | 0.0 | 10.4 | 0.0 | 99.4 | 0.0 | 12.6 |
| | | AIMv2 | 0.0 | 99.0 | 0.0 | 8.4 | 0.0 | 98.4 | 0.0 | 13.4 |
| | | SigLIP 2 | 0.0 | 99.2 | 0.0 | 10.0 | 0.0 | 68.6 | 0.2 | 13.2 |
| | | DINOv3 | 0.2 | 91.8 | 0.0 | 8.8 | 4.6 | 37.6 | 3.0 | 8.0 |
| | | TiTok | 0.0 | 24.4 | 0.0 | 10.0 | 0.2 | 14.6 | 0.0 | 9.8 |
| | | VAR | 0.0 | 24.4 | 0.0 | 10.0 | 0.8 | 14.2 | 0.0 | 8.2 |
| | MoT | CLIP | 0.2 | 94.8 | 0.0 | 11.2 | 7.4 | 99.6 | 5.2 | 18.4 |
| | | AIMv2 | 2.6 | 99.6 | 0.4 | 10.2 | 6.6 | 95.6 | 7.4 | 12.4 |
| | | SigLIP 2 | 0.0 | 99.4 | 0.0 | 10.4 | 0.2 | 80.4 | 0.0 | 8.6 |
| | | DINOv3 | 7.8 | 99.8 | 4.8 | 5.6 | 8.8 | 84.6 | 6.8 | 10.0 |
| | | TiTok | 0.8 | 23.2 | 0.0 | 9.4 | 0.6 | 87.4 | 1.0 | 10.4 |
| | | VAR | 0.0 | 26.8 | 0.0 | 8.0 | 0.0 | 54.0 | 0.0 | 11.2 |
| Grid | Joint-Decoder | CLIP | 4.0 | 87.8 | 1.2 | 18.0 | 0.0 | 58.0 | 0.0 | 21.6 |
| | | AIMv2 | 9.6 | 99.2 | 4.2 | 13.8 | 8.0 | 70.8 | 2.4 | 27.4 |
| | | SigLIP 2 | 1.6 | 99.2 | 1.8 | 18.6 | 6.4 | 73.2 | 3.4 | 17.4 |
| | | DINOv3 | 2.6 | 97.2 | 2.4 | 17.6 | 6.2 | 57.8 | 4.4 | 16.6 |
| | | TiTok | 0.4 | 30.6 | 1.0 | 13.8 | 1.4 | 5.0 | 1.0 | 2.8 |
| | | VAR | 0.4 | 33.2 | 0.0 | 8.0 | 1.2 | 37.0 | 1.2 | 10.0 |
| | Cross Attention | CLIP | 1.4 | 93.8 | 0.0 | 18.4 | 0.0 | 65.6 | 0.0 | 19.2 |
| | | AIMv2 | 0.0 | 91.6 | 0.0 | 13.8 | 0.0 | 15.2 | 0.0 | 8.2 |
| | | SigLIP 2 | 0.0 | 98.6 | 0.0 | 13.2 | 1.2 | 22.4 | 1.4 | 15.0 |
| | | DINOv3 | 2.2 | 34.4 | 2.6 | 11.0 | 1.6 | 32.8 | 1.6 | 10.6 |
| | | TiTok | 0.0 | 30.0 | 0.0 | 11.4 | 4.0 | 16.0 | 1.4 | 8.8 |
| | | VAR | 0.0 | 32.2 | 0.0 | 7.8 | 0.0 | 17.2 | 0.0 | 8.2 |
| | MoT | CLIP | 0.6 | 94.6 | 0.0 | 15.4 | 10.0 | 69.4 | 2.4 | 17.6 |
| | | AIMv2 | 2.0 | 95.4 | 0.2 | 19.0 | 9.6 | 59.2 | 4.2 | 20.8 |
| | | SigLIP 2 | 0.0 | 98.2 | 0.2 | 20.4 | 0.0 | 72.2 | 0.2 | 17.0 |
| | | DINOv3 | 7.4 | 98.0 | 8.2 | 17.2 | 8.2 | 62.4 | 4.6 | 21.6 |
| | | TiTok | 0.2 | 30.8 | 0.2 | 11.4 | 0.0 | 48.4 | 0.0 | 12.4 |
| | | VAR | 0.0 | 30.8 | 0.0 | 10.2 | 0.0 | 40.0 | 0.4 | 10.8 |

**Table 7: Text-to-Image (TQA→VQA) transfer performance (accuracy in %) by different combination of fusion architectures and image tokenizers.** Scores are evaluated after models were fine-tuned on the text-only (TQA) version of the synthetic datasets. Each score is presented alongside the original score before the additional training.

| Dataset | Architecture | Tokenizers | TQA (InD) No Train | TQA (InD) SFT | TQA (OOD) No Train | TQA (OOD) SFT | VQA (InD) No Train | VQA (InD) SFT | VQA (OOD) No Train | VQA (OOD) SFT |
|---|---|---|---|---|---|---|---|---|---|---|
| SpatialMap | Joint-Decoder | CLIP | 22.4 | 74.8 | 22.2 | 74.4 | 24.4 | 25.4 | 24.0 | 27.8 |
| | | AIMv2 | 36.4 | 78.8 | 35.0 | 78.0 | 24.8 | 21.0 | 25.6 | 26.6 |
| | | SigLIP 2 | 38.4 | 75.2 | 40.0 | 77.4 | 24.8 | 0.4 | 25.8 | 0.6 |
| | | DINOv3 | 36.4 | 65.0 | 37.6 | 67.0 | 25.6 | 26.6 | 23.4 | 24.2 |
| | | TiTok | 13.0 | 56.8 | 13.0 | 61.0 | 1.4 | 0.0 | 1.4 | 0.0 |
| | | VAR | 29.8 | 71.6 | 25.4 | 77.6 | 0.0 | 25.4 | 0.2 | 22.8 |
| | Cross Attention | CLIP | 0.2 | 76.6 | 0.0 | 72.8 | 2.6 | 0.0 | 3.0 | 0.0 |
| | | AIMv2 | 6.0 | 62.6 | 4.2 | 65.8 | 21.2 | 0.0 | 23.0 | 0.0 |
| | | SigLIP 2 | 20.6 | 57.2 | 21.2 | 60.8 | 6.8 | 0.0 | 9.0 | 0.0 |
| | | DINOv3 | 7.0 | 61.8 | 5.8 | 63.0 | 1.0 | 0.0 | 0.6 | 0.2 |
| | | TiTok | 26.8 | 64.8 | 27.8 | 64.6 | 25.2 | 0.0 | 25.8 | 0.4 |
| | | VAR | 31.6 | 71.8 | 27.8 | 75.0 | 16.8 | 2.6 | 17.2 | 3.6 |
| | MoT | CLIP | 28.8 | 78.2 | 29.0 | 77.6 | 22.0 | 26.8 | 22.2 | 25.6 |
| | | AIMv2 | 19.4 | 69.4 | 13.8 | 62.6 | 22.4 | 29.0 | 21.4 | 25.2 |
| | | SigLIP 2 | 29.0 | 70.0 | 31.4 | 69.2 | 25.6 | 24.2 | 25.6 | 29.2 |
| | | DINOv3 | 10.2 | 59.6 | 8.8 | 63.4 | 16.2 | 26.6 | 15.6 | 23.0 |
| | | TiTok | 29.2 | 62.2 | 28.0 | 59.8 | 6.6 | 27.2 | 5.2 | 27.0 |
| | | VAR | 26.2 | 70.4 | 28.4 | 72.0 | 5.2 | 24.4 | 9.2 | 22.8 |
| Ring | Joint-Decoder | CLIP | 0.4 | 100.0 | 3.4 | 15.2 | 5.0 | 15.4 | 3.0 | 5.8 |
| | | AIMv2 | 5.6 | 100.0 | 3.4 | 17.2 | 0.8 | 10.8 | 0.0 | 5.6 |
| | | SigLIP 2 | 8.6 | 100.0 | 4.8 | 13.0 | 0.8 | 11.0 | 0.4 | 8.0 |
| | | DINOv3 | 8.2 | 100.0 | 3.4 | 13.8 | 0.4 | 13.0 | 0.0 | 5.0 |
| | | TiTok | 9.0 | 100.0 | 0.0 | 14.8 | 0.6 | 11.2 | 0.0 | 10.4 |
| | | VAR | 3.0 | 100.0 | 3.8 | 17.2 | 0.0 | 9.6 | 0.0 | 6.8 |
| | Cross Attention | CLIP | 0.0 | 100.0 | 0.0 | 9.4 | 0.0 | 10.8 | 0.0 | 5.2 |
| | | AIMv2 | 0.0 | 100.0 | 0.0 | 12.2 | 0.0 | 12.0 | 0.0 | 6.4 |
| | | SigLIP 2 | 0.0 | 100.0 | 0.2 | 14.4 | 0.0 | 1.8 | 0.0 | 0.2 |
| | | DINOv3 | 4.6 | 99.8 | 3.0 | 13.0 | 0.2 | 9.4 | 0.0 | 1.2 |
| | | TiTok | 0.2 | 100.0 | 0.0 | 13.6 | 0.0 | 5.8 | 0.0 | 3.6 |
| | | VAR | 0.8 | 100.0 | 0.0 | 13.6 | 0.0 | 6.6 | 0.0 | 3.6 |
| | MoT | CLIP | 7.4 | 100.0 | 5.2 | 15.6 | 0.2 | 11.2 | 0.0 | 4.8 |
| | | AIMv2 | 6.6 | 100.0 | 7.4 | 11.6 | 2.6 | 6.6 | 0.4 | 4.0 |
| | | SigLIP 2 | 0.2 | 100.0 | 0.0 | 15.0 | 0.0 | 7.6 | 0.0 | 5.8 |
| | | DINOv3 | 8.8 | 100.0 | 6.8 | 14.6 | 7.8 | 5.4 | 4.8 | 4.0 |
| | | TiTok | 0.6 | 100.0 | 1.0 | 14.6 | 0.8 | 7.0 | 0.0 | 3.0 |
| | | VAR | 0.0 | 100.0 | 0.0 | 11.0 | 0.0 | 11.0 | 0.0 | 8.2 |
| Grid | Joint-Decoder | CLIP | 0.0 | 99.8 | 0.0 | 70.6 | 4.0 | 11.6 | 1.2 | 4.0 |
| | | AIMv2 | 8.0 | 99.8 | 2.4 | 29.4 | 9.6 | 11.2 | 4.2 | 7.6 |
| | | SigLIP 2 | 6.4 | 99.8 | 3.4 | 74.8 | 1.6 | 13.0 | 1.8 | 6.0 |
| | | DINOv3 | 6.2 | 100.0 | 4.4 | 63.6 | 2.6 | 14.0 | 2.4 | 6.8 |
| | | TiTok | 1.4 | 100.0 | 1.0 | 36.2 | 0.4 | 14.0 | 1.0 | 8.8 |
| | | VAR | 1.2 | 97.8 | 1.2 | 39.6 | 0.4 | 11.8 | 0.0 | 4.6 |
| | Cross Attention | CLIP | 0.0 | 99.8 | 0.0 | 43.0 | 1.4 | 10.2 | 0.0 | 6.4 |
| | | AIMv2 | 0.0 | 98.2 | 0.0 | 31.6 | 0.0 | 9.6 | 0.0 | 3.8 |
| | | SigLIP 2 | 1.2 | 100.0 | 1.4 | 54.0 | 0.0 | 11.8 | 0.0 | 6.4 |
| | | DINOv3 | 1.6 | 100.0 | 1.6 | 26.8 | 2.2 | 13.0 | 2.6 | 7.0 |
| | | TiTok | 4.0 | 100.0 | 1.4 | 45.4 | 0.0 | 8.6 | 0.0 | 7.2 |
| | | VAR | 0.0 | 99.8 | 0.0 | 25.2 | 0.0 | 8.0 | 0.0 | 3.6 |
| | MoT | CLIP | 10.0 | 100.0 | 2.4 | 55.6 | 0.6 | 9.0 | 0.0 | 5.0 |
| | | AIMv2 | 9.6 | 100.0 | 4.2 | 26.4 | 2.0 | 18.6 | 0.2 | 10.0 |
| | | SigLIP 2 | 0.0 | 100.0 | 0.2 | 47.2 | 0.0 | 11.8 | 0.2 | 5.0 |
| | | DINOv3 | 8.2 | 100.0 | 4.6 | 30.8 | 7.4 | 11.4 | 8.2 | 5.0 |
| | | TiTok | 0.0 | 99.8 | 0.0 | 25.4 | 0.2 | 12.0 | 0.2 | 6.0 |
| | | VAR | 0.0 | 99.8 | 0.4 | 35.2 | 0.0 | 15.2 | 0.0 | 6.2 |

# B ADDITIONAL RESULTS: CROSS-MODALITY TRANSFER FOR OPEN-WEIGHT VLMS

To supplement our analysis on cross-modality and out-of-distribution transfer on our trained vision-language models, in this section we evaluate a suite of open-source vision–language models spanning different architectures and scales, including Qwen2.5-VL (3B- and 7B-Instruct), Gemma, InternVL3, and Kimi-VL. Each model is fine-tuned on our synthetic datasets using supervised fine-tuning (SFT) for direct comparison across modalities. For Qwen2.5-VL, we additionally explore reinforcement learning (RL)-based finetuning, reported in Section B.2, to assess whether optimization beyond standard SFT can further enhance cross-modality transfer. Hyperparameters and other details about training setup are provided in Appendix D.

## B.1 SUPERVISED FINE-TUNING TRANSFER RESULTS

Similar to Section 4.1 and Section 4.2, we perform a single-epoch supervised fine-tuning (SFT) on either the image-based VQA task or the equivalent text-only TQA task across our three synthetic datasets to evaluate cross-modality transfer in open-weight vision–language models. Training is conducted on the respective train split, and models are evaluated on four held-out settings: (i) test split in the same modality, (ii) test split in the opposite modality, (iii) out-of-distribution (OOD) test split in the same modality, and (iv) OOD test split in the opposite modality. This setup parallels the fine-tuning procedure used for our in-house multimodal models, enabling direct comparison of how open-source architectures generalize across modalities and dataset variants.

We present evaluation accuracy after fine-tuning on the image version of the datasets in Table 8. We observe consistent image-to-text transfer when models are fine-tuned on the image version of the datasets. Qwen2.5-VL-7B achieves the strongest overall performance, with InternVL3 also showing competitive transfer, particularly on Grid. As in our earlier experiments, SpatialMap remains the most challenging for cross-modal transfer: several models exhibit drops in text-only accuracy after SFT on the image task, underscoring the modality mismatch in how the task is represented.

Conversely, in the text-to-image transfer setting, results are more mixed, as can be seen in Table 9. Qwen2.5-VL-7B shows strong transfer on SpatialMap but weaker performance on Grid and Ring. InternVL3 demonstrates more balanced transfer across datasets, suggesting that pretraining with substantial text-only data may aid robustness when moving from textual to visual reasoning. Overall, these results reinforce the asymmetric nature of cross-modality transfer and highlight model-specific differences in how supervision in one modality propagates to another.

## B.2 COMPARISON TO REINFORCEMENT LEARNING FINE-TUNING

We compare supervised fine-tuning (SFT) and reinforcement learning (RL) fine-tuning for Qwen-2.5-VL models at two scales (3B and 7B parameters). In both cases, we train exclusively on the image version of each synthetic dataset (SpatialMap, Ring, Grid), using a binary reward signal of 1 for producing the correct final answer and 0 otherwise. We then evaluate on both the in-distribution test and OOD test splits, across both image and text modalities. This setup probes not only in-distribution performance, but also image-to-text cross-modality transfer and robustness to distribution shift. We report results at base (before fine-tuning on synthetic data), after SFT, and at the best RL checkpoint for each configuration.

At the larger 7B scale (see Table 10), RL generally provides stronger generalization than SFT, aligning with similar sentiments from prior work (Chu et al., 2025; Liu et al., 2025; Setlur et al., 2025). For example, RL improves cross-modality transfer on SpatialMap, where SFT hurts image to text performance, and yields large gains on OOD splits for Ring (0.87–0.88 vs 0.14–0.20 for SFT). RL also preserves or slightly enhances in-distribution accuracy, often reaching near-perfect performance. Although there were instances where SFT frequently saturates or even degrades transfer, it is more sample-efficient; we see for certain tasks (eg. Grid), the gains from RL are more modest particularly with respect to cross-modality transfer. However, we observed RL training continued to yield steady improvements for the entire duration of training (15 epochs)— in contrast, SFT offers better sample efficiency.

**Table 8: Image-to-Text (VQA→TQA) transfer performance (accuracy in %).** Models were fine-tuned on the VQA version of the synthetic datasets. Each score is presented alongside its performance delta from the base model's performance (before fine-tuning on synthetic data). For both TQA (InD) and TQA (OOD), the highest final score among the models is bolded to compare performance on cross-modality transfer (VQA→TQA) and difficulty generalization (InD vs OOD). The number in the parentheses indicates the change from base.

| Dataset | Model | VQA (InD) | VQA (OOD) | TQA (InD) | TQA (OOD) |
|---|---|---|---|---|---|
| SpatialMap | Qwen2.5-VL-7B | 98.6 ( +18.4) | 98.2 ( +18.0) | **63.0 ( +0.8)** | 57.6 ( -5.8) |
| | Qwen2.5-VL-3B | 84.4 ( +38.8) | 81.0 ( +38.6) | 42.2 ( -1.4) | 43.0 ( -1.0) |
| | InternVL3-8B | 97.2 ( +19.4) | 95.2 ( +19.4) | 62.0 ( -6.2) | **59.2 ( -9.4)** |
| | Gemma-3-4B | 94.0 ( +53.4) | 91.4 ( +50.0) | 52.8 ( -1.6) | 53.8 ( +1.8) |
| | Kimi-VL-A3B | 28.8 ( +4.0) | 24.8 ( -3.6) | 34.2 ( -24.2) | 28.6 ( -33.8) |
| Grid | Qwen2.5-VL-7B | 99.4 ( +83.2) | 84.8 ( +77.0) | **99.4 ( +47.0)** | **95.6 ( +68.2)** |
| | Qwen2.5-VL-3B | 91.0 ( +79.0) | 34.4 ( +28.6) | 94.4 ( +82.0) | 60.0 ( +53.2) |
| | InternVL3-8B | 99.4 ( +80.2) | 68.4 ( +53.6) | 97.8 ( +11.6) | 87.4 ( +18.4) |
| | Gemma-3-4B | 99.0 ( +83.6) | 59.8 ( +56.0) | 98.2 ( +66.2) | 84.4 ( +71.6) |
| | Kimi-VL-A3B | 7.0 ( -4.4) | 2.6 ( -4.8) | 60.6 ( -2.4) | 44.8 ( -5.6) |
| Ring | Qwen2.5-VL-7B | 99.4 ( +83.8) | 15.0 ( +4.6) | **99.6 ( +69.4)** | 15.8 ( -4.0) |
| | Qwen2.5-VL-3B | 85.0 ( +74.4) | 13.8 ( +5.4) | 82.0 ( +69.8) | 18.2 ( +10.8) |
| | InternVL3-8B | 99.4 ( +84.4) | 17.8 ( +9.4) | 95.4 ( +41.2) | 28.8 ( -40.6) |
| | Gemma-3-4B | 99.8 ( +89.6) | 15.2 ( +12.8) | 99.4 ( +34.6) | 15.2 ( -15.8) |
| | Kimi-VL-A3B | 7.0 ( -3.0) | 3.0 ( -6.6) | 63.4 ( +33.0) | **34.0 ( -0.4)** |

**Table 9: Text-to-Image (TQA→VQA) transfer performance (accuracy in %).** Models were fine-tuned on the text-only (TQA) version of the synthetic datasets. Each score is presented alongside its performance delta from the base model's performance (before fine-tuning on synthetic data). For both VQA (InD) and VQA (OOD), the highest final score among the models is bolded to compare performance on cross-modality transfer (TQA→VQA) and difficulty generalization (InD vs OOD). The number in the parentheses indicates the change from base.

| Dataset | Model | TQA (InD) | TQA (OOD) | VQA (InD) | VQA (OOD) |
|---|---|---|---|---|---|
| SpatialMap | Qwen2.5-VL-7B | 86.2 ( +24.0) | 90.0 ( +26.6) | **89.0 ( +8.8)** | **86.2 ( +6.0)** |
| | Qwen2.5-VL-3B | 61.4 ( +17.8) | 55.0 ( +11.0) | 27.0 ( -18.6) | 27.2 ( -15.2) |
| | InternVL3-8B | 91.4 ( +23.2) | 93.6 ( +25.0) | 78.6 ( +0.8) | 79.2 ( +3.4) |
| | Gemma-3-4B | 90.2 ( +35.8) | 90.4 ( +88.2) | 52.6 ( +12.0) | 48.6 ( +7.2) |
| | Kimi-VL-A3B | 70.2 ( +11.8) | 73.2 ( +10.8) | 22.8 ( -2.0) | 25.4 ( -3.0) |
| Grid | Qwen2.5-VL-7B | 99.8 ( +47.4) | 95.4 ( +68.0) | 19.2 ( +3.0) | 8.2 ( +0.4) |
| | Qwen2.5-VL-3B | 100.0 ( +87.6) | 49.2 ( +42.4) | 16.8 ( +4.8) | 9.8 ( +4.0) |
| | InternVL3-8B | 100.0 ( +13.8) | 98.4 ( +29.4) | 16.0 ( -3.2) | 4.8 ( -10.0) |
| | Gemma-3-4B | 99.8 ( +67.8) | 54.4 ( +41.6) | 16.4 ( +1.0) | **11.2 ( +7.4)** |
| | Kimi-VL-A3B | 90.8 ( +27.8) | 41.4 ( -9.0) | **60.6 ( +49.2)** | 4.0 ( -3.4) |
| Ring | Qwen2.5-VL-7B | 100.0 ( +69.8) | 16.4 ( -3.4) | 18.6 ( +3.0) | 7.0 ( -3.4) |
| | Qwen2.5-VL-3B | 93.2 ( +81.0) | 23.0 ( +15.6) | **21.4 ( +10.8)** | 8.0 ( -0.4) |
| | InternVL3-8B | 99.8 ( +45.6) | 23.6 ( -45.8) | 19.8 ( +4.8) | **9.8 ( +1.4)** |
| | Gemma-3-4B | 99.2 ( +34.4) | 16.2 ( -14.8) | 14.6 ( +4.4) | 9.2 ( +6.8) |
| | Kimi-VL-A3B | 94.0 ( +63.6) | 41.8 ( +7.4) | 10.4 ( +0.4) | 9.0 ( -0.6) |

For the smaller 3B scale (see Table 11), transfer patterns are more varied. RL improves cross-modality and OOD transfer on SpatialMap, but offers less consistent gains on Grid, where SFT remains competitive—likely reflecting that the Grid and Ring tasks have more closely aligned image and text representations. A notable pitfall emerges in Ring: the 3B RL model fails to improve much beyond random chance, collapsing into short outputs with only the final answer token. This illustrates the importance of controlling model output distribution during RL, and the inherent limitations of RLVR to the support of the base model Wu et al. (2025). As mentioned above, when such

**Table 10: Image-to-Text (VQA→TQA) transfer performance (accuracy in %) for Qwen-2.5-VL-7B-Instruct.** Synthetic SpatialMap, Grid, and Ring results at base (before fine-tuning on synthetic data), after SFT, and best RL checkpoint for Qwen-2.5-VL-7B-Instruct when training with the image-based version (VQA) of the respective task. For each test split, we bold the higher score between SFT and RL. The number in the parentheses indicates the change from base.

| | SpatialMap | | | Grid | | | Ring | | |
|---|---|---|---|---|---|---|---|---|---|
| | Base | SFT | RL | Base | SFT | RL | Base | SFT | RL |
| **VQA** (InD) | 80.2 | 99.6 (+19.4) | **99.8 (+19.6)** | 16.2 | **99.8 (+83.6)** | 98.6 (+82.4) | 15.6 | **100.0 (+84.4)** | 99.8 (+84.2) |
| **VQA** (OOD) | 80.2 | 99.6 (+19.4) | **99.8 (+19.6)** | 7.8 | **89.0 (+81.2)** | 59.6 (+51.8) | 10.4 | 15.0 (+4.6) | **86.6 (+76.2)** |
| **TQA** (InD) | 62.2 | 65.2 (+3.0) | **70.8 (+8.6)** | 52.4 | **99.8 (+47.4)** | 61.8 (+9.4) | 30.2 | **100.0 (+69.8)** | 98.6 (+68.4) |
| **TQA** (OOD) | 63.4 | 63.8 (+0.4) | **71.6 (+8.2)** | 27.4 | **97.4 (+70.0)** | 30.4 (+3.0) | 19.8 | 19.8 (+0.0) | **87.8 (+68.0)** |

**Table 11: Image-to-Text (VQA→TQA) transfer performance (accuracy in %) for Qwen-2.5-VL-3B-Instruct.** Synthetic SpatialMap, Grid, and Ring results at base (before fine-tuning on synthetic data), after SFT, and best RL checkpoint for Qwen-2.5-VL-3B-Instruct when training with the image-based version (VQA) of the respective task. For each test split, we bold the higher score between SFT and RL.. The number in the parentheses indicates the change from base.

| | SpatialMap | | | Grid | | | Ring | | |
|---|---|---|---|---|---|---|---|---|---|
| | Base | SFT | RL | Base | SFT | RL | Base | SFT | RL |
| **VQA** (InD) | 45.6 | 99.2 (+53.6) | **100.0 (+54.4)** | 12.0 | **99.2 (+87.2)** | 89.6 (+77.6) | 10.6 | **100.0 (+89.4)** | 13.2 (+2.6) |
| **VQA** (OOD) | 42.4 | **99.4 (+57.0)** | **99.4 (+57.0)** | 5.8 | **45.4 (+39.6)** | 44.8 (+39.0) | 8.4 | **17.4 (+9.0)** | 12.4 (+4.0) |
| **TQA** (InD) | 43.6 | 46.4 (+2.8) | **64.4 (+20.8)** | 12.4 | **99.2 (+86.8)** | 42.0 (+29.6) | 12.2 | **99.6 (+87.4)** | 12.8 (+0.6) |
| **TQA** (OOD) | 44.0 | 44.8 (+0.8) | **66.0 (+22.0)** | 6.8 | **60.0 (+53.2)** | 20.4 (+13.6) | 7.4 | **21.4 (+14.0)** | 9.6 (+2.2) |

collapse does not occur, we observed RL training yields steady improvements over epochs, whereas SFT tends to plateau earlier.

Overall, these results indicate that RL at larger scale consistently enhances generalization across modality and distribution shift, while at smaller scale it can either unlock improved transfer (SpatialMap) or suffer from instability (Ring). SFT remains a useful baseline for more aligned tasks but appears less reliable for tasks requiring substantial abstraction across modalities.

## C SYNTHETIC DATASET DETAILS

Below we provide details regarding the three synthetic datasets.

**SpatialMap:** The Synthetic SpatialMap dataset tests models on spatial reasoning over symbolic objects. Each sample consists of a set of $n$ colored shapes placed randomly on a blank canvas. Questions probe pairwise spatial relations, for example: "Where is the blue circle relative to the red star? Possible options: Northeast, Northwest, Southeast, Southwest." The dataset follows the structure of the spatial mapping tasks in prior work (Wang et al., 2024), with object names replaced by colored geometric shapes for greater visual simplicity and compositional control. We provide two parallel modalities: an image version, where the model must infer relations directly from the visual configuration, and a text version, which encodes equivalent information as a series of binary relation statements (e.g., "The purple pentagon is to the Northeast of the blue circle."). The text description is complete and sufficient to solve the task, but requires chaining relational statements. The training split contains 4,500 procedurally generated examples with 7 objects per image, while the OOD split introduces 8 objects per image, slightly increasing task complexity without changing the basic query format.

**Grid:** The Synthetic Grid dataset evaluates navigation and reasoning in discrete two-dimensional environments. We generate an $n \times n$ grid where each cell contains a distinct colored shape. Questions specify a starting shape and a sequence of relative moves, such as "Begin at the yellow triangle and go down one step, then right two steps. Which shape do you land on?" The image version provides the grid visualization alongside the question, while the text version lists the grid contents in row-major order, thereby encoding object positions without requiring visual perception. This dataset is adapted from the "global grid" task of Yamada et al. (2024), though we substitute ImageNet categories with geometric shapes to simplify object recognition and emphasize spatial reasoning. The training set uses $3 \times 3$ grids with navigation sequences of length 8, while the OOD split increases difficulty with $4 \times 4$ grids and up to 12-step sequences. Unlike the image version, the text version bypasses object recognition, highlighting how modality differences impact reasoning difficulty.

**Ring:** The Synthetic Ring dataset parallels the grid task but in a circular layout, also present in its text version in Yamada et al. (2024). Objects are evenly arranged around a ring, and queries specify a starting shape and a number of clockwise or counterclockwise steps (e.g., "Starting from the red square, move four steps clockwise. Which shape do you land on?"). The image version presents the circle of shapes with the question, while the text version linearizes the ring into an ordered list beginning from a designated reference point and continuing clockwise. The training split contains rings of 8 objects with navigation sequences up to 8 steps, while the OOD split expands to 12 objects and 12-step sequences.

**Dataset generation:** We describe how we procedurally generate the three datasets below. We plan on releasing the dataset generation code and example datasets. When evaluating our models, we provide the question directly as displayed in Figure 3 and append `Output your final answer after writing 'Final answer:' in your response.`

- For each **SpatialMap** example, we sample a set of distinct color-shape pairs and randomly assign them positions along two independent orderings (vertical and horizontal). These orderings determine the relative row and column of each object, which are rendered on a blank canvas. We then select an unambiguous query pair $(q, r)$ whose vertical and horizontal offsets uniquely determine a diagonal relation (Northeast, Northwest, Southeast, or Southwest). The **text modality** question (`question`) begins with a full relational description of the scene expressed as binary statements (e.g., "The purple pentagon is to the Northeast of the blue circle."). The query is appended to this description, requiring the model to chain multiple statements to answer correctly. The **image modality** question (`question_direct`) instead lists only the set of objects without relational information, with the same query appended, such that solving requires interpreting the image directly. Solutions are generated in parallel: the text solution (`solution`) provides a multi-step chain-of-thought reasoning through vertical and horizontal relations, while the image solution (`solution_direct`) gives a concise explanation phrased as direct visual inspection. The final label (`answer`) is the correct diagonal relation.

- For each **Grid** example, we construct an $n \times n$ grid and assign a unique color-shape object to each cell, rendering the grid with light boundaries. A navigation query is created by sampling a start cell and a valid sequence of directional moves. The **text modality** question (`question`) specifies the grid contents in row-major order, ensuring that the layout can be reconstructed entirely from text, followed by the navigation program (e.g., "Start at the yellow triangle, move down one step, then right two steps. Which object do you land on?"). The **image modality** question (`question_direct`) instead lists the objects in random order without positional information, requiring the model to resolve the navigation over the visual grid. The solutions mirror these formats: the text solution (`solution`) details each step of the navigation trace, while the image solution (`solution_direct`) is identical, describing the sequence of moves until the final object is reached. The final label (`answer`) is the object found at the destination cell.

- For each **Ring** example, we arrange a sampled set of unique color–shape objects evenly around a circle in clockwise order, selecting a starting position and a sequence of clockwise or counterclockwise steps to form a navigation query. The **text modality** question (`question`) lists the objects deterministically in clockwise order from a fixed reference point, then provides the navigation program (e.g., "Starting from the red square, move four steps clockwise. Which object do you land on?"). The **image modality** question (`question_direct`) instead lists the same objects in random order without positional information, such that the model must rely on the ring image to resolve the query. Both the text and image solutions (`solution`, `solution_direct`) provide an explicit step-by-step trace of the walk around the ring, concluding with the identified object. The final label (`answer`) is the object at the destination position.

## D  TRAINING DETAILS

### D.1  STAGE 1 (PRETRAINING) HYPERPARAMETERS

For Stage 1 pretraining, models with Qwen3-0.6B as the LLM backbone are trained on the COYO-700M dataset (Byeon et al., 2022) for 100,000 steps with a global batch size of 1536. Input text captions are truncated to a maximum length of 256 tokens. During this stage, only the Joint-Decoder and Cross-Attention models are trained directly. For stage 2, MoT checkpoints are initialized from the resulting Joint-Decoder weights, as mentioned in Section 2.3.

We follow the learning rate strategy of Deitke et al. (2024). The adapter modules for the Joint-Decoder and Cross-Attention models use a learning rate of $2 \times 10^{-4}$. When the image tokenizer is unfrozen, it is trained with a learning rate of $6 \times 10^{-6}$. All models use a cosine-decay learning rate schedule with a linear warmup, decaying to 10% of the maximum learning rate by the end of training. A comprehensive list of all hyperparameters is provided in Table 12.

**Table 12: Hyperparameters for stage 1 training.**

|  | Joint-Decoder | Cross-Attention |
|---|---|---|
| Adapter LR | $2 \times 10^{-4}$ | $2 \times 10^{-4}$ |
| Cross-Attention LR | N/A | $2 \times 10^{-4}$ |
| Image Tokenizer LR | $6 \times 10^{-6}$ | $6 \times 10^{-6}$ |
| Optimizer | AdamW | AdamW |
| Betas | (0.9, 0.999) | (0.9, 0.999) |
| Weight decay | 0.01 | 0.01 |
| LR Schedule | Cosine Decay | Cosine Decay |
| Min LR | 10% of Max | 10% of Max |
| Linear Warmup | 5000 steps | 5000 steps |
| Global Batch size | 1536 | 1536 |
| Num Training Steps | 100k steps | 100k steps |

For the models with a larger backbone, we change the global batch size and number of steps to adjust for our compute resources and the training requirements of our models. In particular, models with a Qwen3-1.7B backbone are trained on COYO-700M with a global batch size of 768 for 50k steps and with a 2500 step warmup. The models with a Qwen3-4B backbone are trained with DataComp-1B (Gadre et al., 2023) with a global batch size of 2048 for 100k steps and a 5000 step warmup. All other hyperparameters are the same as the Qwen3-0.6B runs.

### D.2  STAGE 2 (FINE-TUNING) HYPERPARAMETERS

For the Stage 2 fine-tuning, models are trained on a combined dataset comprising COCO-Captions, VQAv2, ChartQA, TextVQA, and DocVQA. Following the second-stage setup of Deitke et al. (2024), we set the learning rate to $1 \times 10^{-5}$ for the image tokenizer (when unfrozen), the adapters in the Joint-Decoder and MoT models, and the cross-attention layers. A higher learning rate of $5 \times 10^{-5}$ is used for the LLM backbone (when unfrozen) and the image modality-transformer in the MoT models. All models use a cosine-decay learning rate schedule with a linear warmup, decaying to 10% of the maximum learning rate. A complete summary of these hyperparameters is available in Table 13.

### D.3  STAGE 3 (REASONING-TRANSFER) HYPERPARAMETERS

For the Stage 3 reasoning transfer experiments, models are fine-tuned on either the text-only (TQA) or image-based (VQA) training split of one of the three synthetic datasets. A uniform learning rate is applied to all unfrozen layers. As in previous stages, discrete image tokenizers remain frozen to avoid the need for a straight-through estimator.

Although the training was configured for 5 epochs using a cosine-decay schedule, we exclusively use the checkpoint saved after the first epoch for all evaluations. Under this setup, the learning rate

**Table 13: Hyperparameters for stage 2 training.**

|  | Joint-Decoder | Cross-Attention | MoT |
|---|---|---|---|
| Adapter LR | $1 \times 10^{-5}$ | $1 \times 10^{-5}$ | $1 \times 10^{-5}$ |
| Cross-Attention LR | N/A | $1 \times 10^{-5}$ | N/A |
| Image Tokenizer LR | $1 \times 10^{-5}$ | $1 \times 10^{-5}$ | $1 \times 10^{-5}$ |
| Image Transformer LR | N/A | N/A | $5 \times 10^{-5}$ |
| Language LR | $5 \times 10^{-5}$ | $5 \times 10^{-5}$ | $5 \times 10^{-5}$ |
| Optimizer | AdamW | AdamW | AdamW |
| Betas | (0.9, 0.999) | (0.9, 0.999) | (0.9, 0.999) |
| Weight decay | 0.01 | 0.01 | 0.01 |
| LR Schedule | Cosine Decay | Cosine Decay | Cosine Decay |
| Min LR | 10% of Max | 10% of Max | 10% of Max |
| Linear Warmup | 750 steps | 750 steps | 750 steps |
| Global Batch size | 1536 | 1536 | 1536 |
| Num Training Steps | 10860 steps (15 epochs) | 10860 steps (15 epochs) | 10860 steps (15 epochs) |

only decays to approximately 90.5% of its initial value by the end of the first epoch ($\frac{1+\cos(\pi/5)}{2} \approx 0.905$), effectively creating a near-constant learning rate. While we expect performance to be very similar to using a true constant learning rate, we provide these specifics for full reproducibility. All hyperparameters are detailed in Table 14.

**Table 14: Hyperparameters for stage 3 training.**

|  | Joint-Decoder | Cross-Attention | MoT |
|---|---|---|---|
| Learning Rate | $1 \times 10^{-5}$ | $1 \times 10^{-5}$ | $1 \times 10^{-5}$ |
| Optimizer | AdamW | AdamW | AdamW |
| Betas | (0.9, 0.999) | (0.9, 0.999) | (0.9, 0.999) |
| Weight decay | 0.01 | 0.01 | 0.01 |
| LR Schedule | Cosine Decay | Cosine Decay | Cosine Decay |
| Linear Warmup | 0 steps | 0 steps | 0 steps |
| Global Batch size | 32 | 32 | 32 |
| Num Training Steps | 140 steps (1 epoch) | 140 steps (1 epoch) | 140 steps (1 epoch) |

### D.4 OPEN-WEIGHT VISION-LANGUAGE MODELS

For our fine-tuning results on open-weight vision-language models presented in Appendix B, we use the LLaMA-Factory framework (Zheng et al., 2024) for SFT and the verl (Sheng et al., 2024) implementation of Group Relative Policy Optimization (GRPO) (Shao et al., 2024) for RL fine-tuning. We specify hyperparameters in Table 15 below.

|  | SFT | RL |
|---|---|---|
| Learning Rate | $1 \times 10^{-6}$ | $1 \times 10^{-6}$ |
| Optimizer | AdamW | AdamW |
| Betas | (0.9, 0.999) | (0.9, 0.999) |
| Warmup Steps | 0 | 0 |
| Scheduler | Cosine Decay | Constant |
| Num Epochs | 1 | 15 |
| Training Global Batch Size | 64 | 128 |
| Rollout Global Batch Size | N/A | 128 |
| N Samples per Prompt | N/A | 5 |
| KL Coeff | N/A | $1 \times 10^{-3}$ |

**Table 15: Hyperparameters for SFT and RL training.**

# E   LARGE LANGUAGE MODEL USAGE

We used large language models to help improve the clarity and style of the manuscript. All drafts, including the main text, citations, and tables, were originally written by hand. We then used ChatGPT-4.1 and Gemini 2.5 Pro to suggest revisions for clarity, conciseness, and academic tone. No content generation or data analysis was performed by language models; all substantive contributions and data interpretation are the authors' own.

