# OpenReview forum: "Understanding the Design Space and Cross-Modality Transfer for Vision-Language Models"
_ICLR.cc/2026/Conference — Submitted to ICLR 2026_

### Official Review · Reviewer_tpUk · 2025-10-23

**Soundness:** 2
**Presentation:** 2
**Contribution:** 2
**Rating:** 2
**Confidence:** 4

**Summary:**

The paper explores what design choices have a large impact on VLMs' VQA and reasoning performance.
This paper examines image tokenizers, fusion architectures, and layer-freezing strategies for the design choices.
The authors train 50+ model variants on a Qwen3-0.6B backbone and evaluate on standard benchmarks plus three new synthetic datasets designed to probe cross-modality transfer.

**Strengths:**

The paper tackles an interesting question of what steps create good VLMs.

The controlled experimental design effectively isolates individual design choices (tokenizer, architecture, freezing strategy) that are typically confounded in VLM research.

Provides actionable insights for practitioners (e.g., Takeaways 1-8).

**Weaknesses:**

Limited model size. The authors only use a single sub-1B model. This begs the question of whether the results are still valid when scaling up the models. While I understand the authors may be limited in their available compute, the question they seek to answer requires more computational power. Hence, the authors should instead use their compute to explore questions within their realm.

Only single runs. No tests of significance.

Unfair Architectural Comparisons. MoT has 400M+ extra parameters vs Joint-Decoder despite "comparable FLOPs." Table 2 does not normalize for parameter count—MoT's advantage when all frozen may simply be due to more capacity. You should add parameter-matched baselines or explicitly report params/FLOPs for each config.

While the authors explore what setup gives the best performance, they do not tackle the why?

There is also no error analysis.

There is no discussion of risks regarding data contamination. There are datasets from 2014, so models like SigLIP 2 have likely seen samples from it.

“Our results show that the Mixture-of-Transformers (MoT) architecture is particularly effective.” Why do you state this? On what grounds? See under Questions for more details.

The reasoning performance metrics should include what random guessing would yield.

The table captions need to be above the tables as per the formatting instructions: “The table number
and title always appear before the table.”
The table captions are also incomplete and missing, for instance, what the bold numbers represent and what can be drawn from the table.
Table formatting is inconsistent. The style of Tables 1 and 2 is very different from that of Tables 3 to 6.

There are some odd hyperlinks, such as SigLIP 2, on lines 217 and 219. Also, why are the references to Sections, Tables, and Figures in red?

Section 2 is missing a subsection on evaluation metrics.

Given the datasets you generate, you should consider (and cite!) https://arxiv.org/abs/2407.06581. Also, you should include the full prompts given to the models.

The findings, while useful, are largely confirmatory (text supervision helps, unfreezing trades off in/out-domain performance) rather than surprising. The cross-modality transfer analysis, though interesting, is hampered by methodological constraints.

Typos etc.:
Line 115: “which contains of 28 transformer layers” should be “which contains 28 transformer layers”
Line 236: +5.3 is missing colour.

**Questions:**

How do you conclude: “MoT with an unfrozen image tokenizer and frozen language layers delivers the best overall task performance?”
The joint decoder with everything unfrozen delivers 1.9pp higher in distribution performance with 3.2 pp lower out of distribution performance. However, depending on how you weigh the two metrics, the conclusion changes. Furthermore, if I looked at improvement over the baseline frozen model, then the conclusion changes yet again.
Please create a single table extension of Table 2 where you do not aggregate on the tokenizers. (It should include all tokenizers and not just the subset you used for Table 2).

For the start of Section 4, why do you create your own datasets rather than using existing ones?

---

> ### Author Response · Authors · 2025-11-25
>
> We thank the reviewer for their careful review of our work. We are glad they find our motivating central research question an important one, and our controlled experimental design effectively isolates these numerous design choices for VLM training. We address each of their comments and feedback below.
>
> In our revision, we’ve introduced new experiments with the new DINOv3 tokenizer, which is a tokenizer trained on a similar scale as AIMv2 and SigLIP 2 but without text objectives. We’ve also added experiments with LLM backbones with up to 4B parameters, and our scaling results can be seen at the end of Section 3. We’ve also focused on fewer, more central takeaways while adding more clarity and precision to the surrounding exposition. Our changes are indicated by the blue text in the revision.
>
> For the two runs using both DINOv3 and Qwen3-4B, we are still generating the VQAv2 test results, and have used the VQAv2 validation scores for aggregating in-domain accuracy at scale. Based on all of our previous experiments, this would affect the results by a small fraction of a percentage point for the relevant models. This would not change any of the takeaways, and we will update the in-domain accuracy plot when we have the test scores, which we expect to have soon.
>
> ### Weaknesses
>
> > Limited model size. The authors only use a single sub-1B model. This begs the question of whether the results are still valid when scaling up the models. While I understand the authors may be limited in their available compute, the question they seek to answer requires more computational power. Hence, the authors should instead use their compute to explore questions within their realm.
>
> Thank you for this comment, we agree that looking at scaling trends and investigating analogous findings for other VLMs is important. To test other LLMs as well as scaling trends, we ran experiments with Qwen3-1.7B and Qwen3-4B as the LLM backbone for the Joint-Decoder and MoT architectures with SigLIP 2 and the new DINOv3 tokenizer (which has a similar architecture and data training scale as SigLIP 2). We’ve added our results to Section 3. As the model scales, we still find that SigLIP 2 (which has an AR text objective) performs vastly better than DINOv3 (which has no text objectives) for in-domain tasks and generally performs better on out-domain tasks. We also compare the MoT architecture with frozen language layers to the fully unfrozen Joint-Decoder architecture and observe that, at all scales and for both tokenizers, this MoT training recipe has out-of-domain results exceeding the corresponding fully unfrozen Joint-Decoder model without losing too much in-domain performance.
>
> To see if our observations held at scale, we chose this subset of ablations as a computationally-feasible way to compare similar tokenizers with different training objectives and see the performance of MoT with frozen language layers. Please let us know if there are additional experiments you would like to see which would substantially improve the understanding of our takeaways at scale.
>
> > Only single runs. No tests of significance… There is also no error analysis.
>
> Due to the size of the design space we explore—spanning multiple tokenizers, architectural variants, training recipes, and both VQA/TQA directions across several datasets—the full cross-product of configurations is already quite large. Under our available compute budget, it was not feasible to run multiple random seeds for each configuration, and we acknowledge that having statistical error bars would strengthen the empirical claims.
>
> Given the budget constraints, we chose to prioritize coverage across scales rather than depth in the form of repeated runs, as we hope the breadth of consistent trends observed across architectures, modalities, and in-distribution/OOD splits provides complementary evidence for the validity of our conclusions. This study of a wide breadth of single runs also appears frequently in the literature, with some examples listed below.
>
> - Hoffmann, Jordan et al. “Training Compute-Optimal Large Language Models.” arXiv preprint arXiv:2203.15556 (2022).
> - Liang, Weixin et al. “Mixture-of-Transformers: A Sparse and Scalable Architecture for Multi-Modal Foundation Models.” arXiv preprint arXiv:2411.04996 (2024)
> - Qi, Zhenting et al. “EvoLM: In Search of Lost Language Model Training Dynamics.” arXiv preprint arXiv:2506.16029 (2025)

---

> > ### Author Response · Authors · 2025-11-25
> >
> > > Unfair Architectural Comparisons. MoT has 400M+ extra parameters vs Joint-Decoder despite "comparable FLOPs." Table 2 does not normalize for parameter count—MoT's advantage when all frozen may simply be due to more capacity. You should add parameter-matched baselines or explicitly report params/FLOPs for each config.
> >
> > We understand the reviewer’s concerns on potential unfair comparisons among fusion architectures based on parameter count. We use comparable FLOPs as a metric because it is tightly linked to both the training and inference time of a model, which is a major bottleneck as the model scales. Additionally, we assume the setting in which we are already given a pretrained LLM of a fixed size and tasked to build the best multimodal model, or VLM, off of the given LLM. Our goal was to identify the best fusion architecture or training strategy to build a VLM from this given pretrained LLM. Under this setting, the LLM backbone is fixed so it’s hard to do parameter-matched baselines. We have stated and clarified this setting in the new revision of the paper.
> >
> > > While the authors explore what setup gives the best performance, they do not tackle the why?
> >
> > Thank you for your comment. In our revision, we’ve changed the exposition to more clearly explain the reasons for some of the behaviors we observe. We expand on how the MoT architecture achieves strong out-of-domain performance by leveraging the more general text knowledge and reasoning of the base language model, and we add additional figures and experiments with DINOv3 to isolate the impact of text-aware vs. text-blind tokenizers. We do note that deep learning is largely an empirical field (see References below for similar works) and deeper insight, such as understanding the features generated by the various text-aware and text-blind tokenizer objectives and why they are useful for downstream VLM modeling, are largely open questions in the field. We’ve expanded our conclusion to raise these open questions.
> >
> > References:
> > - Shi, Min et al. “Eagle: Exploring the Design Space for Multimodal LLMs with Mixture of Encoders.” arXiv preprint arXiv:2408.15998 (2024).
> > - Tong, Shengbang et al. “Cambrian-1: A Fully Open, Vision-Centric Exploration of Multimodal LLMs.” arXiv preprint arXiv:2406.16860 (2024).
> >
> > > There is no discussion of risks regarding data contamination. There are datasets from 2014, so models like SigLIP 2 have likely seen samples from it.
> >
> > Some of the benchmarks (i.e. VQAv2, DocVQA) have closed versions, requiring a submission to a server to evaluate on a withheld dataset. Additionally, certain benchmarks are newer (e.g. RealWorldQA), having been released as recently as last year. Potential data contamination when training and evaluating foundation models is a general confounder in the broader community, an inherent issue when using other researchers’ models as backbones.
> >
> > > “Our results show that the Mixture-of-Transformers (MoT) architecture is particularly effective.”  Why do you state this? On what grounds? See under Questions for more details.
> >
> > Thank you for your questions and comments. We agree that the best-performing combination could change depending on how one weighs in-domain and out-of-domain performance, and there are various tradeoffs one must consider depending on the expected downstream use cases. We revised Section 3 to be clearer and more precise about these tradeoffs. In particular, we highlight how the MoT architecture is able to get the strongest out-of-domain scores by enabling strategies that preserve the more general knowledge and reasoning of the base LLM, without severely harming in-domain scores. We have also added new scaling experiments at the end of Section 3 to check that this behavior holds as we scale up the model, where we continue to see the MoT architecture get strong out-of-domain performance by freezing the language layers.
> >
> > A de-aggregated table with results for every architecture/tokenizer/freezing combination can be found in Appendix A.1.
> >
> > > The reasoning performance metrics should include what random guessing would yield.
> > The table captions need to be above the tables as per the formatting instructions: “The table number and title always appear before the table.” The table captions are also incomplete and missing, for instance, what the bold numbers represent and what can be drawn from the table. Table formatting is inconsistent. The style of Tables 1 and 2 is very different from that of Tables 3 to 6.
> >
> > Thank you for pointing out the table formatting instructions! We’ve made the appropriate changes in the new version and moved the captions above the tables. We’ve unified the table styles for Tables 1 and 2. We’ve replaced the other tables with figures that convey the main takeaways more effectively. For reasoning performance results, we replaced the tables with new bar charts along with an indicator for the expected random guess score to clearly show the cross modality transfer after reasoning fine-tuning.

---

> > > ### Author Response · Authors · 2025-11-25
> > >
> > > > There are some odd hyperlinks, such as SigLIP 2, on lines 217 and 219. Also, why are the references to Sections, Tables, and Figures in red?
> > >
> > > We couldn’t see the hyperlinks on Lines 217 and 219 as you mentioned (let us know if this still persists in the new uploaded version). The red links were as a result of using `colorlinks’. We find the color helpful for indicating that these are clickable links for navigating to the corresponding section/table/figure. We’ve also seen similar colored links in ICLR papers and in the general literature (some examples listed below) and have found the links in those papers helpful.
> > >
> > > References:
> > > - Hoffmann, Jordan et al. “Training Compute-Optimal Large Language Models.” arXiv preprint arXiv:2203.15556 (2022).
> > > - Yang, Sherry et al. “Learning Interactive Real-World Simulators.” ICLR 2024. https://openreview.net/forum?id=sFyTZEqmUY
> > > - Qi, Xiangyu et al. “Safety Alignment Should be Made More Than Just a Few Tokens Deep” ICLR 2025. https://openreview.net/forum?id=6Mxhg9PtDE
> > >
> > > > Section 2 is missing a subsection on evaluation metrics.
> > >
> > > In subsection 2.4, we have a list of all of the evaluation metrics we use, along with short descriptions of what each evaluation tests. If there is particular information you were looking for in this subsection, please let us know so we can address this in the revised version.
> > >
> > > > Given the datasets you generate, you should consider (and cite!) https://arxiv.org/abs/2407.06581. Also, you should include the full prompts given to the models.
> > > The findings, while useful, are largely confirmatory (text supervision helps, unfreezing trades off in/out-domain performance) rather than surprising. The cross-modality transfer analysis, though interesting, is hampered by methodological constraints.
> > >
> > > Thank you for this suggestion. We have added the citation to Section 4 and Related Works. We have also clarified our prompting setup in Appendix C: for all models, we simply present the questions in the format illustrated in Figure 3 and append the instruction “Output your final answer after writing ‘Final answer:’ in your response.” In addition, we will open-source all synthetic datasets, generation scripts, and training/evaluation code to make our setup fully transparent and reproducible.
> > >
> > > Regarding the comment about the methodological constraints of our cross-modality transfer analysis, we would be grateful for clarification. We currently provide full details of our data generation pipeline and perturbation procedures in Appendix C, as well as complete finetuning configurations for our pretrained models in Appendix D.3 and for open models in Appendix D.4, with both SFT and RL results summarized in Appendix B. From our perspective, these choices were designed to make the transfer experiments as controlled and interpretable as possible. If there are specific methodological limitations you have in mind (e.g., additional ablations, controls, or reporting that would make the analysis stronger), we would appreciate further guidance so we can address them in the revised version.
> > >
> > > > Typos etc.: Line 115: “which contains of 28 transformer layers” should be “which contains 28 transformer layers” Line 236: +5.3 is missing colour.
> > >
> > > Thank you for identifying these typos, we have edited them in the new version.

---

> > > > ### Author Response · Authors · 2025-11-25
> > > >
> > > > ### Questions
> > > >
> > > > > How do you conclude: “MoT with an unfrozen image tokenizer and frozen language layers delivers the best overall task performance?”
> > > >
> > > > See the previous point on MoT being effective.
> > > >
> > > > > For the start of Section 4, why do you create your own datasets rather than using existing ones?
> > > >
> > > > We agree it is natural to ask why we do not directly reuse existing benchmarks, and have revised the beginning of Section 4 to make our motivation more explicit and explain how current datasets do not meet our need for what we wanted to test, hence why we generated our own. Our goal in this section is to systematically study cross-modality transfer in a small number of tightly controlled settings, where (i) image–text pairs contain exactly the same information, (ii) we can create matched InD/OOD splits, and (iii) the visual input is as simple as possible so that we can cleanly separate perception from spatial reasoning. While our setup is inspired by prior work, the existing datasets we cite are not directly suited to this analysis. In [1], the “map” tasks only include test splits (no training data for fine-tuning open-weight models), do not explicitly probe out-of-distribution generalization by increasing the number of objects, and use procedurally generated natural-language place names (e.g., “Himalayan Hot Springs”) that are also rendered as text in the image, which introduces confounds beyond basic spatial relations. In [2], the tasks are evaluated only in text form on LLMs—there are no image-based counterparts—and the labels are drawn from ImageNet categories rather than a small set of colored shapes. For our purposes, we therefore opted to generate new paired image/text datasets built around simple colored shapes, with matched train/test and OOD splits, so that we can directly and systematically measure cross-modality transfer under controlled conditions.
> > > >
> > > > [1] Wang, Jiayu, et al. "Is a picture worth a thousand words? delving into spatial reasoning for vision language models." Advances in Neural Information Processing Systems 37 (2024): 75392-75421
> > > >
> > > > [2] Yamada, Yutaro, et al. "Evaluating spatial understanding of large language models." arXiv preprint arXiv:2310.14540 (2023).

---

> > ### Author Response · Authors · 2025-12-03
> >
> > We've updated the paper with the test-dev scores for VQAv2. As expected, the final in-domain averages for the two DINOv3 runs using the Qwen3-4B backbone barely changed (both increased about 0.3 points), so the results still support our takeaways.

---

### Official Review · Reviewer_HxG6 · 2025-10-30

**Soundness:** 2
**Presentation:** 3
**Contribution:** 2
**Rating:** 4
**Confidence:** 2

**Summary:**

This paper broadly evaluates existing vision-language model approaches, covering image tokenizers, model architecture, and frozen parameter settings.
Each method is evaluated on several vision question answering datasets in terms of several perspectives including in-domain/out-of-domain settings and cross-modalirty transfer from image/text to text/image.
Through experimental evaluations, this paper provides some observations, e.g., the best approach for image tokenizers is to learn with text-alignment objectives.

**Strengths:**

- Readers easily grasp the claims of this paper.
The experimental evaluation setup is clearly organized. In addition, the conclusion of each experiment is well summarized in takeaways.

- Broadly and fairly evaluating existing technologies provides useful information for practitioners.
This paper provides a comprehensive evaluation of key aspects in handling vision language models, including continuous/discrete image tokenizers and frozen/unfrozen approaches,
from several perspectives including in-domain and out-of-domain scenarios.

- Model architecture and image tokenizers are crucial elements in vision language models.
If the paper had provided a clearer and more promising research direction, it could have had a significant impact.
However, as noted in Weakness, the claims are not sufficiently supported by the experimental evidence.

**Weaknesses:**

- In the evaluation, the language backbone consists solely of one model, Qwen3-0.6B. This might cause some bias in experiments although I do not know the impact of language backbone.

- The experimental evaluation mainly serves as a benchmark comparison across existing techniques and concludes without deeper evaluation based on hypotheses or analytical questions.
Since no experiments were designed to substantiate the observations, the takeaways might not go beyond observations.
 Of course, benchmark evaluations are important, but the paper becomes stronger by presenting analytical experiments that go beyond observations,
 identifying open questions that should be addressed as a field, and outlining future directions for vision language models.

- Regarding the above issue, I'm not sure if the conclusion in the takeaways is correct.
Takeaway 1 claims "Image tokenizers trained with text-alignment objectives are crucial for strong VLM performance.".
Its basis is that image tokenizers trained with text-alignment objectives (AIMv2, SigLIP 2, CLIP) outperform those trained for image reconstruction (TiTok, VAR).
However, in Section 2.2, the distinction between AIMv2, SigLIP 2, CLIP and TiTok, VAR is whether they are continuous tokenizers or discrete tokenizers.
Thus, it is also possible to conclude that continuous tokenizers outperform discrete tokenizers.
Takeaway 5 makes a similar claim regarding cross-modality transfer. However, since Table 3 and Table 1 show similar trends, it is also possible that TiTok and VAR simply have lower performance.

**Questions:**

If there are any misunderstandings on my part in Weakness, could you point them out?

---

> ### Author Response · Authors · 2025-11-25
>
> We thank the reviewer for their thoughtful comments and are encouraged that they found our claims easy to grasp, the experimental setup organized, and our broad evaluation of key VLM components useful for practitioners. We address their comments regarding scale and consolidating our takeaways below.
>
> In our revision, we’ve introduced new experiments with the new DINOv3 tokenizer, which is a tokenizer trained on a similar scale as AIMv2 and SigLIP 2 but without text objectives. We’ve also added experiments with LLM backbones with up to 4B parameters, and our scaling results can be seen at the end of Section 3. We’ve also focused on fewer, more central takeaways while adding more clarity and precision to the surrounding exposition. Our changes are indicated by the blue text in the revision.
>
> For the two runs using both DINOv3 and Qwen3-4B, we are still generating the VQAv2 test results, and have used the VQAv2 validation scores for aggregating in-domain accuracy at scale. Based on all of our previous experiments, this would affect the results by a small fraction of a percentage point for the relevant models. This would not change any of the takeaways, and we will update the in-domain accuracy plot when we have the test scores, which we expect to have soon.
>
> ### Weaknesses
>
> > In the evaluation, the language backbone consists solely of one model, Qwen3-0.6B. This might cause some bias in experiments although I do not know the impact of language backbone.
>
> Thank you for this comment, we agree that looking at scaling trends and investigating analogous findings for other VLMs is important. To test other LLMs as well as scaling trends, we ran experiments with Qwen3-1.7B and Qwen3-4B as the LLM backbone for the Joint-Decoder and MoT architectures with SigLIP 2 and the new DINOv3 tokenizer (which has a similar architecture and data training scale as SigLIP 2). We’ve added our results to Section 3. As the model scales, we still find that SigLIP 2 (which has an AR text objective) performs vastly better than DINOv3 (which has no text objectives) for in-domain tasks and generally performs better on out-domain tasks. We also compare the MoT architecture with frozen language layers to the fully unfrozen Joint-Decoder architecture and observe that, at all scales and for both tokenizers, this MoT training recipe has out-of-domain results exceeding the corresponding fully unfrozen Joint-Decoder model without losing too much in-domain performance.
>
> To see if our observations held at scale, we chose this subset of ablations as a computationally-feasible way to compare similar tokenizers with different training objectives and see the performance of MoT with frozen language layers. Please let us know if there are additional experiments you would like to see which would substantially improve the understanding of our takeaways at scale.
>
> > The experimental evaluation mainly serves as a benchmark comparison across existing techniques and concludes without deeper evaluation based on hypotheses or analytical questions. Since no experiments were designed to substantiate the observations, the takeaways might not go beyond observations. Of course, benchmark evaluations are important, but the paper becomes stronger by presenting analytical experiments that go beyond observations, identifying open questions that should be addressed as a field, and outlining future directions for vision language models.
>
> Thank you for this comment. As noted in our response to Reviewer Y52k, we have revised the paper to consolidate the contributions into three main takeaways, and we agree that the work should go beyond reporting benchmark numbers. Our experiments were designed to be analytical rather than purely descriptive—for example, our systematic ablations over image tokenizer, fusion architecture, and, in particular, freezing strategies across both stages directly probe how full LLM fine-tuning versus partial unfreezing affects in-domain performance, out-of-distribution robustness, and cross-modality transfer.
>
> In the revised conclusion, we now make the resulting actionable directions explicit: (i) understanding why full LLM fine-tuning often harms OOD robustness and visual grounding (e.g., catastrophic forgetting or misalignment with frozen vision components), (ii) designing new training objectives and curricula that explicitly promote cross-modality alignment and transfer (such as cross-modal consistency losses or structured intermediate representations), and (iii) testing whether the training recipe we identify in Section 3 and the observed transfer asymmetries persist at larger scales and in other modality pairs (e.g., speech–text). We hope these clarifications make it clearer that our empirical study is intended to surface concrete questions for VLM architecture and training design.

---

> > ### Author Response · Authors · 2025-11-25
> >
> > > Regarding the above issue, I'm not sure if the conclusion in the takeaways is correct. Takeaway 1 claims "Image tokenizers trained with text-alignment objectives are crucial for strong VLM performance.". Its basis is that image tokenizers trained with text-alignment objectives (AIMv2, SigLIP 2, CLIP) outperform those trained for image reconstruction (TiTok, VAR). However, in Section 2.2, the distinction between AIMv2, SigLIP 2, CLIP and TiTok, VAR is whether they are continuous tokenizers or discrete tokenizers. Thus, it is also possible to conclude that continuous tokenizers outperform discrete tokenizers. Takeaway 5 makes a similar claim regarding cross-modality transfer. However, since Table 3 and Table 1 show similar trends, it is also possible that TiTok and VAR simply have lower performance.
> >
> > Thank you for this comment. We agree that there are confounding factors since all of the generative tokenizers are discrete and the text-understanding tokenizers are continuous. We’ve added experiments for DINOv3, which is a continuous tokenizer trained without any text objectives at a similar data scale to SigLIP 2 and AIMv2, and we do find that DINOv3 generally performs worse than these two text-aligned (now referred to as “text-aware” in the revision) tokenizers on our downstream academic benchmarks. We’ve recategorized the tokenizers in Section 2 to split between text-aware tokenizers, which are trained with some text-related objective, and text-blind tokenizers, which have no such text objective. We’ve also consolidated and rewritten our takeaways to be more precise as to the impact of the different types of tokenizers. We hope these experiments clarify the impact of text-aware objectives on downstream performance.

---

> > > ### Comment · Reviewer_HxG6 · 2025-11-28
> > >
> > > Thank you for your response!
> > > I am also grateful for your efforts, such as conducting additional experiments.
> > > However, considering that this paper is exploratory research investigating hypotheses and conclusions experimentally,
> > > the changes made in the revision seem major modifications.
> > > Re-evaluating the revision within a limited timeframe is difficult, and I think that re-submission for re-review is appropriate.
> > > I hope that a more comprehensive investigation and refinement will make the paper stronger.
> > >
> > > Finally, I would like to share a new question about the response.
> > > Regarding the new model, while it is limited to a scale change, might models other than Qwen be viable?

---

> > > > ### Author Response · Authors · 2025-12-03
> > > >
> > > > >Thank you for your response! I am also grateful for your efforts, such as conducting additional experiments. However, considering that this paper is exploratory research investigating hypotheses and conclusions experimentally, the changes made in the revision seem major modifications.
> > > >
> > > > Our changes address the concerns of all of the reviewers by adding additional ablations along with better presentation and clarity. We have not made any major modifications to the main thesis of the work, but instead have strengthened our takeaways with more empirical evidence.
> > > >
> > > > > Regarding the new model, while it is limited to a scale change, might models other than Qwen be viable?
> > > >
> > > > Our experimental framework and code (which we plan to publicly release upon acceptance) is compatible with other LLMs with slight modifications. We note that other related papers like Cambrian-1 [1], Eagle [2], and NVLM [3] also use a single LLM backbone at each scale for their ablations.
> > > >
> > > >  [1] Tong et al. “Cambrian-1: A Fully Open, Vision-Centric Exploration of Multimodal LLMs.”      Advances in Neural Information Processing Systems 37 (2024): 87310--87356.
> > > >
> > > > [2] Shi et al. “Eagle: Exploring The Design Space for Multimodal LLMs with Mixture of Encoders.” ICLR 2025.
> > > >
> > > > [3] Dai et al. “NVLM: Open Frontier-Class Multimodal LLMs.” arXiv preprint arXiv:2409.11402 (2024).

---

> > ### Author Response · Authors · 2025-12-03
> >
> > We've updated the paper with the test-dev scores for VQAv2. As expected, the final in-domain averages for the two DINOv3 runs using the Qwen3-4B backbone barely changed (both increased about 0.3 points), so the results still support our takeaways.

---

### Official Review · Reviewer_Y52k · 2025-10-31

**Soundness:** 2
**Presentation:** 3
**Contribution:** 2
**Rating:** 4
**Confidence:** 4

**Summary:**

This paper presents a systematic empirical study of the design space for vision-language models (VLMs). The authors investigate three key dimensions: image tokenization, architectural fusion mechanisms, and layer-freezing strategies. By training and evaluating over 50 VLM variants built on a Qwen3-0.6B backbone, the paper derives some takeaways about what design choices lead to better performance on in-domain and out-of-domain tasks.

**Strengths:**

The paper is exceptionally well-written and organized.

The experimental methodology is comprehensive.

**Weaknesses:**

1. All experiments are conducted on a 0.6B parameter LLM backbone. The paper's takeaways are presented as general principles for VLM design, but without validation on larger, more capable models, they may simply be artifacts of a low-capacity regime.

2. Several of the main takeaways are well-known, such as takeaway 1 & 2 & 3.

3. I am not sure how to get a certain conclusion of this paper, it seems more like to be a blog & survey.

**Questions:**

I'm curious what the author's highest priority takeaway is.

---

> ### Author Response · Authors · 2025-11-25
>
> We thank the reviewer for taking the time to review our work and are glad they found the paper well-written and organized. We address their comments regarding scale and consolidating our takeaways below.
>
> In our revision, we’ve introduced new experiments with the new DINOv3 tokenizer, which is a tokenizer trained on a similar scale as AIMv2 and SigLIP 2 but without text objectives. We’ve also added experiments with LLM backbones with up to 4B parameters, and our scaling results can be seen at the end of Section 3. We’ve also focused on fewer, more central takeaways while adding more clarity and precision to the surrounding exposition. Our changes are indicated by the blue text in the revision.
>
> For the two runs using both DINOv3 and Qwen3-4B, we are still generating the VQAv2 test results, and have used the VQAv2 validation scores for aggregating in-domain accuracy at scale. Based on all of our previous experiments, this would affect the results by a small fraction of a percentage point for the relevant models. This would not change any of the takeaways, and we will update the in-domain accuracy plot when we have the test scores, which we expect to have soon.
>
> ### Weaknesses
>
> > All experiments are conducted on a 0.6B parameter LLM backbone. The paper's takeaways are presented as general principles for VLM design, but without validation on larger, more capable models, they may simply be artifacts of a low-capacity regime.
>
> Thank you for this comment, we agree that looking at scaling trends and investigating analogous findings for other VLMs is important. To test other LLMs as well as scaling trends, we ran experiments with Qwen3-1.7B and Qwen3-4B as the LLM backbone for the Joint-Decoder and MoT architectures with SigLIP 2 and the new DINOv3 tokenizer (which has a similar architecture and data training scale as SigLIP 2). We’ve added our results to Section 3. As the model scales, we still find that SigLIP 2 (which has an AR text objective) performs vastly better than DINOv3 (which has no text objectives) for in-domain tasks and generally performs better on out-domain tasks. We also compare the MoT architecture with frozen language layers to the fully unfrozen Joint-Decoder architecture and observe that, at all scales and for both tokenizers, this MoT training recipe has out-of-domain results exceeding the corresponding fully unfrozen Joint-Decoder model without losing too much in-domain performance.
>
> To see if our observations held at scale, we chose this subset of ablations as a computationally-feasible way to compare similar tokenizers with different training objectives and see the performance of MoT with frozen language layers. Please let us know if there are additional experiments you would like to see which would substantially improve the understanding of our takeaways at scale.
>
> > Several of the main takeaways are well-known, such as takeaway 1 & 2 & 3…I am not sure how to get a certain conclusion of this paper, it seems more like to be a blog & survey.
>
> Thank you for this feedback. We agree that the initial version of the paper tried to communicate too many separate takeaways, which made the core contributions harder to see and may have contributed to the impression of a “blog & survey” rather than a focused experimental study. In the revised version, we have consolidated the paper around three central takeaways: in our first two takeaways, across our ablations, we identify a practical recipe in which pairing text-aligned (now referred to as “text-aware” in the revision) image tokenizers with a Mixture-of-Transformers fusion architecture yields strong out-of-domain performance while preserving in-domain and text-only behavior. For our last takeaway, our controlled study of cross-modality transfer on synthetic spatial reasoning tasks shows that transfer is generally weak unless image and text share tightly aligned, structured representations, with image-to-text transfer being relatively strong while we were unable to observe text-to-image transfer.
>
> To make this focus clearer, we (i) removed the earlier list of four conclusions and rewrote Sections 3-4 to point back to these three main conclusions, and (ii) rewrote the conclusion to restate these takeaways and highlight concrete follow-up directions (e.g., understanding why full LLM fine-tuning harms OOD robustness and how to design better cross-modality alignment objectives). We also emphasize that the paper is not just a survey of existing work: beyond discussing design choices, our core contribution is an extensive empirical study with new controlled synthetic datasets and systematic ablations, including additional experiments on open-source VLMs in Appendix B that present analogous cross-modality transfer results.

---

> > ### Author Response · Authors · 2025-11-25
> >
> > ### Questions
> >
> > > I'm curious what the author's highest priority takeaway is.
> >
> > While the first two takeaways are consistent with trends hinted at in prior work, we view our highest priority takeaway as the third and final one. We believe this points to a more fundamental open problem: how to design architectures and training procedures that preserve or improve cross-modality alignment under continued fine-tuning, especially when the fine-tuning tasks are not perfectly matched across modalities. The revised conclusion now explicitly emphasizes this direction and frames our ablations as a starting point for understanding and improving cross-modality transfer.

---

> > ### Author Response · Authors · 2025-12-03
> >
> > We've updated the paper with the test-dev scores for VQAv2. As expected, the final in-domain averages for the two DINOv3 runs using the Qwen3-4B backbone barely changed (both increased about 0.3 points), so the results still support our takeaways.

---

### Official Review · Reviewer_XpAR · 2025-11-01

**Soundness:** 3
**Presentation:** 2
**Contribution:** 3
**Rating:** 4
**Confidence:** 3

**Summary:**

This paper presents a detailed survey of VLM design axes, focusing on image tokenizers, modality fusion architectures, and LLM backbone freezing strategies by altering many different configurations based on a single model (Qwen3-0.6B). Through a consistent three-stage training pipeline (pretraining, finetuning, and cross-modality reasoning transfer), the authors evaluate performance on both in-domain and out-of-domain benchmarks. They report several interesting findings regarding the impact of image tokenizer design, the effect of thawing the language backbone, and comparisons among modality fusion strategies. These results provide actionable insights, although their exclusive use of a very small sized model leaves some room for doubt about the generalization of the findings.

**Strengths:**

- The paper conducts a comprehensive and well-structured examination of previously underexplored VLM design axes (such as image tokenizers, fusion architectures, and layer-freezing strategies) within a consistent framework.

- Their findings yield several actionable insights, which benefits future model development, e.g., the benefit of training image tokenizers with text-alignment objectives, the importance of selecting unfreezing strategies based on target tasks, and the effectiveness of the Mixture-of-Transformers (MoT) architecture.

- In addition, the analysis of cross-modality transfer offers a valuable perspective that goes beyond conventional, often superficial, VQA style benchmarking.

**Weaknesses:**

- All experiments are conducted on a very small 0.6B-parameter model (Qwen3-0.6B), which limits the overall impact of the work. A study of this depth and architectural scope would ideally include experiments on larger models. If training all variants is not feasible, testing a few representative configurations with larger models would help validate the trends observed with the 0.6B model.

- While the paper mentions training "over 50 variants", these are essentially factorial combinations of a few design axes rather than truly distinct model architectures. The framing could better reflect that this is rather an extensive ablation study rather than a broad model comparison.

- Over the years, I have seen several survey papers on the design space of multimodal LLMs (some of these papers are listed below), some of which might already have covered similar architectural and training aspects. These works would be valuable reference points, but the current related work section does not discuss them.

- (Minor) The presentation could be improved. Some results are reported in aggregated form without a clear enumeration of all configurations. For example, the claim of “50+ variants” could be shown more explicitly through a summary table or schematic.

Long, Siqu, et al. "Vision-and-language pretrained models: A survey." arXiv preprint arXiv:2204.07356 (2022).

Du, Yifan, et al. "A survey of vision-language pre-trained models." arXiv preprint arXiv:2202.10936 (2022).

Yin, Shukang, et al. "A survey on multimodal large language models." National Science Review 11.12 (2024): nwae403.

Zhang, Duzhen, et al. "Mm-llms: Recent advances in multimodal large language models." arXiv preprint arXiv:2401.13601 (2024).

Ma, Xiaorui, Haoran Xie, and S. Joe Qin. "Efficiently Integrate Large Language Models with visual perception: A survey from the training paradigm perspective." Information Fusion (2025): 103419.

**Questions:**

See the above weaknesses.

---

> ### Author Response · Authors · 2025-11-25
>
> We thank the reviewer for their careful review of our work and are encouraged that they found our exploration of VLM design axes within a unified framework comprehensive and useful. We appreciate their recognition of the actionable insights on text-aligned (now referred to as “text-aware” in the revision) image tokenizers and our cross-modality transfer analysis. We address each of their comments below.
>
> In our revision, we’ve introduced new experiments with the new DINOv3 tokenizer, which is a tokenizer trained on a similar scale as AIMv2 and SigLIP 2 but without text objectives. We’ve also added experiments with LLM backbones with up to 4B parameters, and our scaling results can be seen at the end of Section 3. We’ve also focused on fewer, more central takeaways while adding more clarity and precision to the surrounding exposition. Our changes are indicated by the blue text in the revision.
>
> For the two runs using both DINOv3 and Qwen3-4B, we are still generating the VQAv2 test results, and have used the VQAv2 validation scores for aggregating in-domain accuracy at scale. Based on all of our previous experiments, this would affect the results by a small fraction of a percentage point for the relevant models. This would not change any of the takeaways, and we will update the in-domain accuracy plot when we have the test scores, which we expect to have soon.
>
> ### Weaknesses
>
> > All experiments are conducted on a very small 0.6B-parameter model (Qwen3-0.6B), which limits the overall impact of the work. A study of this depth and architectural scope would ideally include experiments on larger models. If training all variants is not feasible, testing a few representative configurations with larger models would help validate the trends observed with the 0.6B model.
>
> Thank you for this comment, we agree that looking at scaling trends and investigating analogous findings for other VLMs is important. To test other LLMs as well as scaling trends, we ran experiments with Qwen3-1.7B and Qwen3-4B as the LLM backbone for the Joint-Decoder and MoT architectures with SigLIP 2 and the new DINOv3 tokenizer (which has a similar architecture and data training scale as SigLIP 2). We’ve added our results to Section 3. As the model scales, we still find that SigLIP 2 (which has an AR text objective) performs vastly better than DINOv3 (which has no text objectives) for in-domain tasks and generally performs better on out-domain tasks. We also compare the MoT architecture with frozen language layers to the fully unfrozen Joint-Decoder architecture and observe that, at all scales and for both tokenizers, this MoT training recipe has out-of-domain results exceeding the corresponding fully unfrozen Joint-Decoder model without losing too much in-domain performance.
>
> To see if our observations held at scale, we chose this subset of ablations as a computationally-feasible way to compare similar tokenizers with different training objectives and see the performance of MoT with frozen language layers. Please let us know if there are additional experiments you would like to see which would substantially improve the understanding of our takeaways at scale.
>
>
> > While the paper mentions training "over 50 variants", these are essentially factorial combinations of a few design axes rather than truly distinct model architectures. The framing could better reflect that this is rather an extensive ablation study rather than a broad model comparison.
>
> We agree that referring to our ablations as training `50 variants` is misleading. In the new version we’ve replaced this with explicitly enumerating the design axes (eg. six tokenizers, three architectural variants) and referring to the variants obtained from the cross product as `VLM configurations`.

---

> > ### Author Response · Authors · 2025-11-25
> >
> > > Over the years, I have seen several survey papers on the design space of multimodal LLMs (some of these papers are listed below), some of which might already have covered similar architectural and training aspects. These works would be valuable reference points, but the current related work section does not discuss them.
> >
> > Thank you for highlighting these survey works on the design space of multimodal LLMs; we have added representative references to the Related Works section in the revised version. These papers provide valuable taxonomies and high-level overviews of existing architectures and training paradigms. Our contribution is complementary: we conduct thorough, controlled experiments that vary tokenizers, fusion architectures, and freezing strategies under a shared backbone and training setup. We report updated results including the recent DINOv3 tokenizer not covered in existing surveys and report results at new scales. We additionally analyze cross-modality transfer behavior across all of these ablations and also extend our cross-modality transfer analysis results to open-sourced VLMs and conduct both SFT and RL experiments in Appendix B. Together, this positions our work as an experiment-driven study that presents concrete, up-to-date takeaways in the design space.
> >
> > > (Minor) The presentation could be improved. Some results are reported in aggregated form without a clear enumeration of all configurations. For example, the claim of “50+ variants” could be shown more explicitly through a summary table or schematic.
> >
> > Thank you for this point. Our design space spans multiple axes (e.g., tokenizers, architectural variants, and which components are trainable), and in the main text we chose to aggregate over some of these axes in order to highlight overall trends for a single factor at a time—for example, Table 2 reports averages over tokenizers to isolate the effect of the training recipe. While we believe this view is useful for understanding high-level trends, we agree that it can make it harder to see the full set of underlying configurations that leads to the “50+ variants” claim. We also acknowledge that some of the existing tables are dense and have improved the visual presentation of the cross-modality transfer results in Section 4: we added new bar plots that track VQA (respectively TQA) test performance before and after fine-tuning on the TQA (respectively VQA) version of each task, with random-chance performance marked as a dotted horizontal line. We still retain comprehensive, non-aggregated results in Appendix A for readers who wish to inspect all configurations in detail. If the reviewer has specific preferences for additional presentation changes, we would be happy to incorporate them.

---

> > ### Author Response · Authors · 2025-12-03
> >
> > We've updated the paper with the test-dev scores for VQAv2. As expected, the final in-domain averages for the two DINOv3 runs using the Qwen3-4B backbone barely changed (both increased about 0.3 points), so the results still support our takeaways.

---

### Author Response · Authors · 2025-12-03

We’d like to thank the reviewers and the AC for reviewing our paper, especially given the unusual rebuttal period for this ICLR. We’ve addressed all of the questions and concerns of our reviewers with additional ablations to strengthen our takeaways with more empirical evidence and with better presentation and clarity for our exposition. We highlight and summarize the changes we’ve made below. All of our changes are indicated by blue text in the revised paper.

## DINOv3

To address a reviewer concern about distinguishing the effect of continuous vs discrete tokenizers from the effect of the tokenizer training objective, we’ve added ablations with the new DINOv3 tokenizer (see updated **Table 1**), adding 18 new configurations using the Qwen3-0.6B backbone along with scaling ablations with Qwen3-1.7B and Qwen3-4B (more on scaling below). Because DINOv3 has a similar architecture and scale of data, these experiments emphasize the effect of text-aware vs text-blind objectives and reduce confounding factors like tokenizer quantization layers or availability of data.

## Scaling

A common question among the reviewers concerned how our observations held as the model scaled in parameter count. To test the scaling behavior of our models and confirm that our takeaways hold with increasing model size, we ran experiments with Qwen3-1.7B and Qwen3-4B as the LLM backbone for the Joint-Decoder and MoT architectures with SigLIP 2 and DINOv3. As shown in **Figure 2**, as the model scales, we still find that SigLIP 2 (which has an AR text objective) performs vastly better than DINOv3 (which has no text objectives) for in-domain tasks and generally performs better on out-domain tasks. We also compare the MoT architecture with frozen language layers to the fully unfrozen Joint-Decoder architecture and observe that, at all scales and for both tokenizers, this MoT training recipe has out-of-domain results exceeding the corresponding fully unfrozen Joint-Decoder model without losing too much in-domain performance.

## Consolidated Framing

We’ve restructured the paper around three central takeaways (see **Takeaway 1, Section 3.1**, **Takeaway 2, Section 3.2**, and **Takeaway 3, Section 4.2**) , focusing on the importance of text-aware objectives for tokenizers, the out-of-domain performance of the MoT architecture, and the role of representational alignment for cross-modality transfer.

We also emphasize that our contribution is complementary to the existing literature. As opposed to surveys which provide taxonomies and high-level overviews of existing architectures and training paradigms, we conduct thorough, controlled experiments that vary tokenizers, fusion architectures, and freezing strategies under a shared backbone and training setup. We report updated results including the recent DINOv3 tokenizer not covered in existing literature and report results at new scales. We additionally analyze cross-modality transfer behavior across all of these ablations with a new, synthetic dataset and also extend our cross-modality transfer analysis results to open-sourced VLMs and conduct both SFT and RL experiments in **Appendix B**. Together, this positions our work as an experiment-driven study that presents concrete, up-to-date takeaways in the design space.

## Presentation and Clarity

We revised the exposition to add clarity and cleaner presentation. These changes are listed below:

- Included bar plots with random-guess baselines for reasoning tasks to indicate cross-modality transfer (**Figure 4** and **Figure 5**)
- Tightened prose in the Introduction and Conclusion to tie each result back to one of the three main takeaways
- Added justification for the new synthetic datasets in Section 4 (matched image/text content, controlled InD/OOD splits, simple visuals)
- Unified table formatting with captions above tables
- Fixed hyperlinks and typos

---

### Meta-Review · Area_Chair_9MTP · 2026-01-07

**Summary:**

This paper presents a systematic empirical study of the design space for vision-language models (VLMs), exploring three key dimensions: image tokenization, architectural fusion mechanisms, and layer-freezing strategies. While the topic is relevant, reviewers raised significant concerns:

- Limited scale: Experiments conducted only on 0.6B models, making conclusions potentially unreliable for larger-scale VLMs commonly used in practice.
- Lack of depth: The study remains descriptive without deeper evaluation based on hypotheses or analytical questions to explain why certain design choices work better.
- Limited novelty in findings: Several main takeaways (e.g., text-aware tokenizers improve performance, freezing preserves textual knowledge) are already well-known in the community.

The authors made substantial revisions to address these concerns. However, the extent of modifications is significant enough that the revised paper essentially requires a new round of review to properly assess.
The paper needs re-review given the major revisions. We encourage resubmission to a future venue. Reject.

**Reviewer Concerns:**

Addressed concerns:
- Limited model scale: All experiments conducted only on a single 0.6B model (Qwen3-0.6B), raising concerns about whether findings generalize to larger, more capable VLMs.
- Lack of analytical depth: The study remains descriptive benchmark comparisons without hypothesis-driven experiments to explain why certain designs work better. No error analysis provided.
- Methodological issues: Unfair architectural comparisons (MoT has 400M+ extra parameters vs Joint-Decoder); single runs without significance tests; potential confounds between continuous/discrete tokenizers and text-alignment objectives.
- Missing context: Insufficient discussion of related survey papers on VLM design space; no data contamination analysis for older datasets.
- Presentation issues: Inconsistent table formatting; incomplete captions; odd hyperlinks; results reported in aggregated form without clear enumeration of all configurations.

Outstanding concerns
- Well-known takeaways: Several conclusions (e.g., text-aligned tokenizers help, freezing preserves textual knowledge) are already established in the community, limiting novelty.

**Reviewer Scores:**

The initial ratings for this paper were 4, 4, 4, 2, and no reviewer indicated willingness to raise their score.

---

### Decision · Program_Chairs · 2026-01-26

Reject